# A dynamic ammonia emission model and the online coupling with WRF-Chem (WRF-SoilN-Chem v1.0): development and regional evaluation in China

Chuanhua Ren[1], Xin Huang[1,2], Tengyu Liu[1,2], Yu Song[3], Zhang Wen[4], Xuejun Liu[4], Aijun Ding[1,2], Tong Zhu[3]

[1] School of Atmospheric Sciences, Nanjing University, Nanjing, China
[2] Jiangsu Provincial Collaborative Innovation Center for Climate Change, Nanjing, China
[3] College of Environmental Sciences and Engineering, Peking University, Beijing, China
[4] College of Resources and Environmental Sciences, China Agricultural University, Beijing, China

*Correspondence to*: Xin Huang (xinhuang@nju.edu.cn)

**Abstract.**

Volatilization of ammonia ($NH_3$) from fertilizer application and livestock wastes is an overwhelmingly important pathway of nitrogen losses in agricultural ecosystems and constitutes the largest source of atmospheric $NH_3$. The volatilization of $NH_3$ highly depends on environmental and meteorological conditions, however, this phenomenon is poorly described in current emission inventory and atmospheric models. Here, we develop a dynamic $NH_3$ emission model capable of calculating $NH_3$ emission rate interactively with time- and spatial-varying meteorological and soil conditions. The $NH_3$ flux parameterization relies on several meteorological factors and anthropogenic activity including fertilizer application, livestock waste, traffic, residential and industrial sectors. The model is then embedded into a regional WRF-Chem model and is evaluated against field measurements of $NH_3$ concentrations and emission flux, and satellite retrievals of column loading. The evaluation shows a substantial improvement in the model performance of $NH_3$ flux and ambient concentration in China. The model well represents the spatial and temporal variations of ambient $NH_3$ concentration, indicating the highest emission in the North China Plain (NCP) and Sichuan Basin, especially during summertime. Compared with normal simulations using fixed emission inventory input, this model features superior capability in simulating $NH_3$ emission flux and concentration during drastic weather changes like frontal activities and precipitation. Such advances in emission quantification also improve the model performance of secondary inorganic aerosol on synoptic scales. While more laboratory and field measurements are still needed for better parameterization of $NH_3$ volatilization, the seamless coupling of soil emission with meteorology provides a better understanding of $NH_3$ emission evolution and its contribution to atmospheric chemistry.

## 1 Introduction

Ammonia ($NH_3$) is the most important alkaline gas in the atmosphere and has important impacts on the ecological environment and human health (Behera et al., 2013). Gas-phase $NH_3$ can react with ambient sulfuric and nitric acids to form ammonium

sulfate ($(NH_4)_2SO_4$), ammonium bisulfate ($NH_4HSO_4$) and ammonium nitrate ($NH_4NO_3$) aerosols (Wang et al., 2013), which constitute a significant fraction of atmospheric fine particles associated with potential human health impacts (Emmanouil et al., 2017; Oprea et al., 2017). Besides, soil $NH_3$ volatilization releases a large amount of nitrogen into the atmosphere, which is in turn deposited back to terrestrial and aquatic ecosystems, contributing to acid deposition and eutrophication. Thus, the atmospheric emission, transport and deposition of $NH_3$ play a societally and ecologically important role in the global nitrogen cycle (Fowler et al., 2013; Liu et al., 2022).

Due to the health and ecological significance of atmospheric $NH_3$, a range of air quality models have been applied to investigate its spatiotemporal variation and sink (Spindler et al., 2001; Asman, 2001; Van Pul et al., 2009). To accurately simulate $NH_3$ in numerical models, considerable approaches have been developed to estimate $NH_3$ emissions from natural and anthropogenic sources, which vary greatly in their complexity and data requirements. A common method named "Bottom-up" method is to use emission factors multiplied by activity data (e.g., fertilizer application amounts corresponding to each crop, mileage of motor vehicles, etc.). Based on this approach, several organizations and researchers established gridded $NH_3$ emission inventories, such as the MEIC, PKU-$NH_3$, EDGAR, and REAS (Li et al., 2017; Huang et al., 2012; Paulot et al., 2014; Crippa et al., 2020; Kurokawa and Ohara, 2020). Based on the human activity level and land-use type in different regions, the bottom-up emission inventories can reflect geographic variations. In addition, these inventories offer highly formatting gridded dataset of monthly variation and spatial distribution of $NH_3$ emission, which are extensively utilized in atmospheric chemical transport models.

On both global and regional scales, $NH_3$ is mostly emitted from agricultural activities, mainly including the fertilization and livestock waste (Bouwman et al., 1997). Multiple lines of evidence from field trials, meta-analysis, and a statistical model indeed showed that local meteorological conditions strongly influence ammonia emission rate (Paerl, 2002; Misselbrook et al., 2005; Bouwman et al., 2002). Temperature is the most important meteorological parameter that affects the partial pressure of $NH_3$ in soil by changing the equilibrium constant of the $NH_4^+$(soil)-$NH_3$(soil)-$NH_3$(gas) equilibrium reaction (Van Slyke and Cullen, 1914; Clay et al., 1990). Furthermore, high temperature increases the rate of urea hydrolysis and the diffusion rate of $NH_4^+$ and $NH_3$ in soil (Overrein and Moe, 1967). Riddick et al. (2016) showed that increasing the ground temperature from 290 to 300 K (at a pH of 7) increases the $NH_3$ emissions by a factor of 3. Besides temperature, soil moisture also strongly controls $NH_3$ loss by influencing urea hydrolysis and $NH_3/NH_4^+$ concentration in soil solution. Results from a field experiment showed that the rate of ammonia volatilization was highly limited at low soil moisture even though the ammoniacal N concentration and pH were high (Smith et al., 1988). On the contrary, if the soil moisture content is quite high, the concentration of ammonia in solution tends to be diluted and thus the ammonia volatilization is prone to be reduced (Fenn and Kissel, 1976). Moderate amounts of soil moisture are found to be more favourable for $NH_3$ volatilization. Additionally, rainfall, wind speed and RH have also been proved by laboratory experiment to affect $NH_3$ emissions (Longhini et al., 2020; Parker et al., 2005). Generally, the environmental elements appreciably influencing ammonia emissions have drastic weather-scale variations or diurnal variations. As one of the main monsoon regions, with the most intense agriculture activities in the world, east Asia is a region experiencing complex synoptic weather and high-level $NH_3$ emission (Ding et al., 2017; Van Damme et al., 2015;

Zhang et al., 2017). Dramatic changes of ammonia emissions caused by different meteorological conditions could also substantially influence the level of nitrogen-containing aerosol (e.g. ammonium nitrate) at the same time. However, the majority of current temporal resolution of existing ammonia inventory still remains at the monthly or annual scale, which is

not capable of accurately reflecting the time-resolved and spatial-varying ammonia emission due to weather change on synoptic scale.

Although in recent years, bidirectional flux models that consider the meteorological influence have been applied in regional chemistry models (Cooter et al., 2010; Zhang et al., 2010a), such as the WRF-CMAQ-EPIC model (Pleim et al., 2019), this method requires users to collect and build complex agriculture-related files. In this work, we develop and evaluate a new user-

friendly ammonia flux module (WRF-SoilN-Chem) capable of simulating dynamic $NH_3$ volatilization under different meteorological conditions. The whole $NH_3$ flux model and input dataset are embedded directly into the WRF-Chem model and can be activated by simply turning on an option in the model control file. It then enables the seamless coupling of meteorology simulation, the $NH_3$ flux flows and atmospheric chemistry module. The WRF-SoilN-Chem model is easy to install and use, open source, version controlled and well documented. In this paper, section 2 presents the overall methodology

including a detailed description of the model's framework, data source and emission factor algorithm. Section 3 compares the simulated $NH_3$ flux and concentrations with site measurements and satellite retrievals, and in-depth analyses of case studies where the simulation of ammonia and secondary inorganic aerosol are improved on a synoptic scale. Discussion of model uncertainty and future improvement of the model like bidirectional parameterization and other dynamic reactive nitrogen emission models are provided in section 4.

## 2 Materials and methods

### 2.1 The parent model: WRF-Chem

WRF-Chem is a state-of-art online coupled meteorology-chemistry model that can simulate meteorological fields and atmospheric chemical compositions including aerosols (Grell et al., 2005). It has been widely used in air quality forecasting and aerosol-related studies (Chen et al., 2016). WRF model provides users with many options for model configurations and

physical schemes, and is used to simulate meteorological processes and advection of atmospheric constituents. WRF uses the Advanced Research WRF (ARW) dynamical solver, which solves fully compressible, Eulerian non-hydrostatic equations on either hybrid sigma–eta (default) or terrain-following vertical coordinates defined by the user. Besides, the WRF model offers many options for land surface physics, planetary boundary layer physics, radiative transfer, cloud microphysics, and cumulus parameterization, for use in meteorological studies, real-time numerical weather prediction, idealized simulations, and data

assimilation on meso- to regional scales (Skamarock, 2019). WRF-Chem is an extended version of WRF including chemical transformation of trace gases and aerosols simultaneously with meteorology. The chemical transport model numerically solves the concentration of chemical species through emissions, advection, vertical mixing with dry deposition, convective transport, gas chemistry, cloud chemistry (for activated aerosols in cloud water), aerosol chemistry, and wet scavenging. WRF-Chem

can simulate trace gases and particles in an interactive way, allowing for feedbacks between the meteorology and radiatively active gases and particles. The detailed model configuration is described in Sect. 3.1.

## 2.2 General framework of Model WRF-SoilN-Chem

The dynamic $NH_3$ flux model within WRF-Chem model estimates the emission rate of $NH_3$ (mol km$^{-2}$ hour$^{-1}$) from the natural and anthropogenic source into the atmosphere at a specific location and time. Figure 1 gives an architectural overview of the WRF-SoilN-Chem coupled model. Briefly, the model contains three parts: (1) the static input data on basic $NH_3$ emissions, (2) the WRF mesoscale meteorological model, (3) and online $NH_3$ emission model coupled into Chem model. The static input data cover the whole China region with ~1 km$^2$ spatial resolution. The emission sources are classified into six sectors, including fertilizer application, livestock waste, agricultural soil, transport, residential and industrial sectors. The data are embedded into *geog_data_path* as binary format and read by *geogrid.exe* in WPS. The WRF model is used to set up the simulation initialization and perform dynamical and physical calculations to get the meteorological parameters for online emission calculation. The online $NH_3$ model is merged into Chem module by modifying the *chem_driver* and *emissions_driver* modules, to simulate $NH_3$ flux under different meteorological conditions.

Users can turn on the dynamic $NH_3$ emission model in WRF-Chem by specifying *nh₃emis_opt = 1* in *namelist.input* control file, similar to the way that users specify the dust emission mechanism in WRF-Chem. The simulated conditions like meteorological element and soil properties provided by WRF solver are transported to $NH_3$ emission model to calculate the meteorology-dependent correction factor (CF). Consequently, the CF is multiplied by the part (1) basic emission data to obtain meteorology-dependent $NH_3$ emission flux. In Chem section, the flux will be considered as source of $NH_3$ in atmosphere, and participate in the next atmospheric physicochemical processes (deposition, accumulation, convection, boundary layer mixing, and chemistry). At the end of simulation, WRF-SoilN-Chem outputs all meteorological parameters, $NH_3$ emission rates and other chemical diagnostic quantities in WRF's standard format.

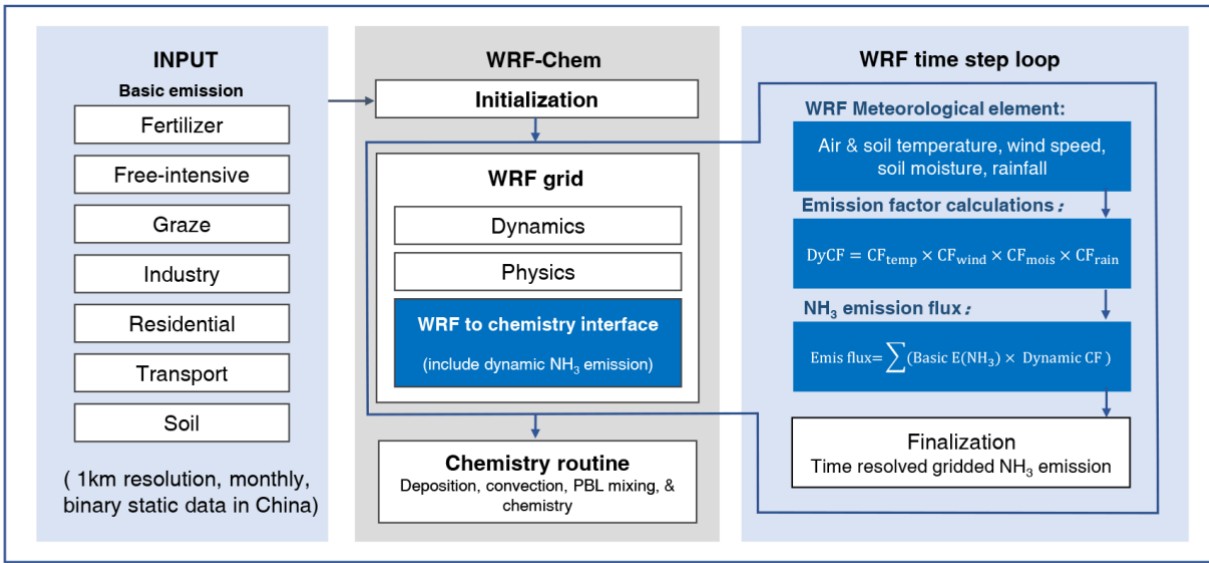

**Figure 1:** Architectural overview of the WRF-SoilN-Chem model (v1.0). The NH$_3$ emission flux calculation (all parts shown in blue) includes basic emission data and dynamic calculation. The parent model WRF-Chem (shown in grey) are standard codes downloaded from their sources, without any modification.

### 2.3 Basic emission data

In this study, the basic emission data used as static input were divided into six sections which are fertilizer application, livestock waste, agricultural soil, transport, residential and industry, covering a total of more than 50 emission sources. Due to intensive agriculture activities in China, synthetic fertilizer application and livestock manure represent the most important sources of

NH$_3$, jointly accounting for more than 80−90 % of total emissions in China (Li et al., 2021). In fertilizer section of basic emission, urea and ammonium bicarbonate (ABC) are two dominant emitters, followed by others like ammonium nitrate and ammonium sulfate. For the livestock waste section, nitrogen in animal excrement in the form of urea can rapidly hydrolyze to form ammonium carbonate and then volatilize as gaseous NH$_3$. The other minor sources include agricultural soil, N-fixing plants, the compost of crop residues, biomass burning, excrement waste from rural populations and chemical industry, waste

disposal, traffic sources and NH$_3$ escape from thermal power plants. All the basic static emissions data were monthly and were obtained by multiplying the monthly activity data and corresponding static EFs, as shown in equation (1). The province-level activity data of important source were obtained from National Bureau of Statistics of China (NBSC), the more detailed information of each source could be found in Table S1.

$$Basic\ E(NH_3) = \sum_i \sum_p \sum_m (A_{i,p,m} \times EF_{static\,i,p,m}) \tag{1}$$

where $Basic\ E_{(NH_3)}$ is the basic NH₃ emissions of a specific source section; $i$, $p$, and $m$ represent source type, the province of China, and the month, respectively. $A_{i,p,m}$ is the activity data of a specific source; and $EF_{static_{i,p,m}}$ is the static emission factor for specific emission sources.

In the fertilizer application section, the fertilizer type, soil pH, fertilizer application rate and method are introduced as parameters to develop EFs for specific conditions. The fertilization rate and method are relatively stable in a month based on the farmers' traditional growing habits. As for soil pH, although it significantly increases after fertilizer application, it gradually fall back to normal state within 30 days due to the nitrification of NH₄⁺ (Curtin et al., 2020). Thus, the pH, fertilizer rate and method were assumed to be relatively stable in monthly scale and were introduced as stable parameters to adjust emission factors for static conditions. The static EF for fertilizer at a specific condition is shown in equation (2):

$$EF_{static_{fertilizer}} = EF_{0i} \times CF_{pH} \times CF_{method} \times CF_{rate} \tag{2}$$

where $EF_{0i}$ is the reference emission factor for a type $i$ fertilizer. $CF_{pH}$ is the correction factor for different soil acidity. $CF_{method}$ is the correction factor for the fertilization method, including basal dressing and top dressing. $CF_{rate}$ is the correction factor for different application rates. The EF₀ for urea and ABC were based on experiments carried out in Henan and Jiangsu Province through the micrometeorological method (Cai et al., 1986; Zhu et al., 1989). The EF₀ for other less prevalent fertilizers refers to the up-to-date and reliable EFs provided by the European Environment Agency (EEA, 2019), as shown in Table S2. The values of $CF_{pH}, CF_{method}, CF_{rate}$ are all referred to Huang et al. (2012) (Table S2).

As for the livestock waste section, free-range, grazing and intensive are three main animal-rearing system in China and produce a huge number of wastes. The NH₃ emissions can be converted into gaseous NH₃ or lost through other pathways during different processes of manure management, including animal housing, manure storage, manure spreading and the grazing stage (Webb and Misselbrook, 2004; Webb et al., 2006). The ammonia emissions from each stage of livestock manure management are affected by many factors, such as species, age, housing structure, manure storage system, spreading technique, time spent outside or inside and meteorological conditions (Zhang et al., 2010b). In the livestock section, the static emission factors are set separately according to the animal species, age, manure status and rearing system, as shown in equation (3):

$$EF_{static_{livestock}} = EF_{c,a,s,f} \tag{3}$$

where $EF_{c,a,m,f}$ represents the emission factor of livestock waste, $c$ represents each animal classes, $a$ represents different age, $s$ represents different rearing system, such as grazing, free-range and intensive, and $f$ represents the form of manures like slurry or solid manures. The typical animal categories and EFs in the estimation of ammonia emission from livestock waste management are listed in Table S3.

As for other miscellaneous sources, further details on the estimation methods and gridded and monthly allocation of the various sources are fully presented in Huang et al. (2012). All the basic emissions are allocated to each 1 km spatial resolution on the basis of land cover, rural population, and other proxies. Eventually, all the data are converted to the binary format and embedded into $geog\_data\_path$ required by WPS.

## 2.4 Meteorology-dependant NH$_3$ dynamic emission factor

In this section, we describe the algorithm for dynamic ammonia emissions from fertilizer application and livestock waste. Many dynamically changing meteorological factors have been proved impacts NH$_3$ emissions significantly. Based on previous works, Tian et al. (2001) showed that near-surface air temperature, soil moisture, wind speed and precipitation these four meteorological factors had the greatest influence on soil ammonia emissions. Besides, Table S4 illustrates these factors have also been required as important meteorological factors in previous ammonia emission models. Therefore, we chose these four meteorological parameters as the main factors modulating emission rate in the parameterization scheme.

For fertilizer application, the main factors governing ammonia volatilization are identified to be ammoniacal N concentration, soil pH, temperature in soil or floodwater, wind speed, soil moisture and rainfall by a variety of lab-experiments. Among these factors, ammoniacal N concentration and soil pH are assumed to be stable in monthly scale and used in monthly basic emission data mentioned above. Other factors that change dynamically under different meteorological conditions need to be calculated in real time in the model. The effects of these dynamical factors on ammonia emissions are reflected by the following correction factors (CFs), and the calculations are performed in WRF-Chem model at each time step.

For fertilization source, the dynamic NH$_3$ emission flux from fertilizer is estimated using Eq. (4):

$$Flux_{NH3-fertilizer} = Basic\ E(NH_3)_{fertilizer} \times CF_{wind} \times CF_{soil_T} \times CF_{soil_m} \times CF_{rain} \tag{4}$$

where $Basic\ E(NH_3)_{fertilizer}$ represents the basic emission presented on Sect. 2.3. $CF_{wind}$ is the correction factor for wind speed; and $CF_{soil_T}$ is the correction factor for soil temperature; $CF_{soil_m}$ is the correction factor for soil moisture content; $CF_{rain}$ is the correction factor for rainfall. The detailed calculation of these $CFs$ is presented in the following sub sections, and Table 1 summarizes the relevant variables used in WRF-SoilN-Chem.

For livestock waste, we divided the NH$_3$ emissions into three sections based on where the manure was located, including manure storage, outdoor and housing. As for the manure storage section, 77% of the manure will be used for composting, 23% will be used for biogas production (Jia, 2014). Biogas is often placed in sealed tanks due to the need for an anaerobic environment. As for composting, the handle of manure site should be "anti-seepage, rain-proof, anti-spillage" (Ministry of Agriculture and Rural Affairs, 2019), so people usually lay fine soil and straw on the ground, and spread a layer of mud or plastic sheeting on the manure or just compost the manure in a closed greenhouse, so as to form a closed environment to avoid the influence of external temperature, wind speed and precipitation. Thus, we assume that NH$_3$ emissions from storage section is not affected by the outdoor environment and therefore does not need to be corrected.

For outdoor farming such as grazing and free range and the application of manure into a field, the excreta were directly deposited in the open air without any treatment. Since the emissions are directly affected by local atmospheric conditions, the estimation of outdoor emission is the same as above for fertilization.

$$Flux_{NH3-hus-outdoor} = Basic\ (E_{outdoor}) \times CF_{wind} \times CF_{soil_T} \times CF_{soil_m} \times CF_{rain} \tag{5}$$

where $Basic\ (E_{outdoor})$ represents the basic emission from outdoor livestock waste, including outdoor grazing in daytime, outdoor free-ranging and manure spreading onto field.

For the housing section, since animals are farmed in buildings, $NH_3$ emissions are directly affected by indoor temperature and ventilation rate. To maintain a healthy environment inside the animal houses a suitable ventilation rate and temperature are required. Specifically, when the outside temperatures drop below a certain level, farmers usually install manual heating to maintain stable temperatures to prevent the decline of animal production. When the outside temperature reaches a maximum level, the mechanical ventilation system is opened to maintain the temperature inside the animal house close to the recommended temperature (Gyldenkaerne et al., 2005). To keep the house clean and animals comfortable, the floor of farmhouse is often with holes or slits to allow the leakage of manure onto the soil below, making the manure easy to be swept away by cleaning machines. So, the manure is still in touch with the soil and therefore will be affected by surface soil moisture.

$$Flux_{NH3-hus-house} = Basic\ (E_{house}) \times CF_{houseV} \times CF_{houseT} \times CF_{soil_m} \tag{6}$$

where $Basic\ (E_{house})$ represents the basic emission from animal houses, including indoor grazing at night, and intensive rearing. The indoor wind speed and temperature are calculated according to outdoor weather conditions, and the detailed calculation methods were reported by (Gyldenkaerne et al., 2005).

### 2.4.1 Wind speed

Increasing wind speed increases the rate of ammonia volatilization by promoting the rapid transport of ammonia away from the surface. Denmead et al. (1982) suggested that the enhanced volatilization at higher wind speeds is due to better mechanical mixing of the N solution in the soil, which replenishes ammonia in the layer next to the surface. And Denmead et al. (1982) found an exponential relationship between the transfer velocity for ammonia and wind speed. After testing and comparing several published approaches, we followed the approach of Gyldenkaerne et al. (2005) to introduce the effects of wind speed on $NH_3$ volatilization from synthetic fertilizer and manure application. The correction factor of wind speed is:

$$CF_{wind} = e^{(0.0419 \times WS)} \tag{7}$$

where $WS$ is the surface wind speed (m s$^{-1}$).

### 2.4.2 Soil temperature

Temperature is the most important meteorological parameter that affects the partial pressure of $NH_3$ in soil by changing the equilibrium constant of the $NH_4^+$(soil)-$NH_3$(soil)-$NH_3$(gas) equilibrium reaction. Several studies have noted an increase in emission from N fertilizers and livestock waste with increasing temperature, and revealed the empirical volatilization rates as functions of air temperatures (Sommer et al., 1991; Pedersen et al., 2021). Mcinnes et al. (1986) reported that diurnal patterns of $NH_3$ loss coincided with fluctuations in soil temperature and water content. However, after our tests, we found that the published relationship can hardly reflect the diurnal pattern of the ammonia flux. Thus, the effects of soil temperature and in situ measurements of $NH_3$ flux conducted by our research group in a typical cropland were involved to derive the $CF$ for synthetic fertilizer and manure emissions in this study. The deriving method and fitting results are demonstrated in Fig. S1. The exponential fitting of $CF_{soil_T}$ is:

$$CF_{soil_T} = e^{(0.093 \times \Delta soil_T - 0.97 + 0.018 \times soil_T)} \tag{8}$$

where $\Delta soil_T$ is the soil temperature gradient, and it can be represented by the difference between the soil temperature of 5 cm and surface skin temperature (K); $soil_T$ is the soil temperature (K) at the depth of 5 cm.

### 2.4.3 Soil water

The soil moisture content is also an important factor controlling ammonia loss. The presence of water in soil is a prerequisite for dissolution of fertilizer N as well as for the hydrolysis of urea, leading to the conversion of other forms N to ammoniacal N in soil solution. Ammonia volatilization can also be enhanced where water evaporates from the soil surface. On the other hand, if the soil moisture content is high, the concentration of $NH_4^+$ in solution will be low due to dilution and $NH_3$ volatilization should be reduced. Ferguson and Kissel (1986) and Fenn and Kissel (1976) also found that ammonia loss was low when soil
moisture was at the extremes. Experiments from Maru et al. (2019) showed that the moderate amounts of soil moisture content were around 50 %, where the $NH_3$ emission reached a peak at this turning point (Maru et al., 2019). Considering the complex effect of soil moisture, we followed the approach of Lian et al. (2021) to introduce the two kinds of effects on $NH_3$ volatilization. The correction factor of soil moisture is:

$$CF_{soil_m} = \begin{cases} 0.45 \times e^{(-1 \times soil_m)} + 0.55 & if \ soil_m \geq 0.5 \\ 0.49 \times e^{soil_m} & if \ soil_m < 0.5 \end{cases} \tag{9}$$

where $soil_m$ represents the soil water content ($m^3 \, m^{-3}$) at the depth of 5 cm.

### 2.4.4 Rain fall

As $NH_3$ readily dissolves in water, $NH_3$ flux can be scavenged by raindrops near the surface (Delitsky and Baines, 2016; Shimshock and De Pena, 1989). Several studies reported that rainfall events after fertilizer application can influence the maximum potential emission of $NH_3$ in the field (Parker et al., 2005; Smith et al., 2009). Wind tunnel experiments from Sanz-
Cobena et al. (2011) showed that the addition of 7 and 14 mm of water to the soil, immediately after urea fertilizing, reduced $NH_3$ emission by 77 % and 89 %, respectively. We derived a relationship between emission rate and rainfall for agriculture emissions based on the available native experimental data (Longhini et al., 2020). The correction factor of rainfall is:

$$CF_{rain} = 1/(3.2 \times rainfall + 1) \tag{10}$$

where $rainfall$ is the precipitation (mm hr$^{-1}$) simulated from the WRF model.


**Table 1. Meteorological and activity variables required to drive WRF-SoilN-Chem**

| No. | Variables in WRF-SoilN-Chem [units] | Description | Usage |
|-----|-------------------------------------|-------------|-------|
| 1. | AGRISOIL [kg km$^{-2}$ month$^{-1}$] | Soil emission | Basic emission |
| 2. | FERTILIZER [kg km$^{-2}$ month$^{-1}$] | Fertilizer emission | Basic emission |
| 3. | FREE-INTEN [kg km$^{-2}$ month$^{-1}$] | Free range & intensive emission | Basic emission |

| | | | |
|---|---|---|---|
| 4. | GRAZE [kg km$^{-2}$ month$^{-1}$] | Grazing emission | Basic emission |
| 5. | INDUSTRY [kg km$^{-2}$ month$^{-1}$] | Industry emission | Basic emission |
| 6. | RESIDENTIAL [kg km$^{-2}$ month$^{-1}$] | Residential emission | Basic emission |
| 7. | TRANSPORT [kg km$^{-2}$ month$^{-1}$] | Transport emission | Basic emission |
| 8. | EFnh3 [unitless] | Dynamic EF | Met-dependent factor |
| 9. | U10 [m s$^{-1}$] | East-west wind at 10m height | Wind factor |
| 10. | V10 [m s$^{-1}$] | North-south wind at 10m height | Wind factor |
| 11. | T2 [K] | Surface temperature | Temperature factor |
| 12. | TSK [K] | Surface skin temperature | Temperature factor |
| 13. | TSLB [K] | Soil temperature | Temperature factor |
| 14. | SMOIS [m$^3$ m$^{-3}$] | Soil moisture | Soil water factor |
| 15. | RAINNC [mm hr$^{-1}$] | Accumulated total grid scale precipitation | Rainfall factor |
| 16. | T_house | Indoor temperature | Temperature factor |
| 17. | V_house | Indoor ventilation rate | Wind factor |
| 18. | freq_residential [unitless] | Residential diurnal pattern | Activity time |
| 19. | freq_industry [unitless] | Industry diurnal pattern | Activity time |
| 20. | freq_transport [unitless] | Transport diurnal pattern | Activity time |
| 21. | rho_phy [kg m$^{-3}$] | the air density | Unit conversion |
| 22. | dtstep [s] | the meteorology big time step in seconds | Time loop |
| 23. | dz8w [m] | the vertical grid spacing for the lowest layer | Unit conversion |
| 24. | emis_ant [mol km$^{-2}$ hr$^{-1}$] | Anthro emission rate | Total emission |
| 25. | current_hour | Emission time | Time |

## 2.5 Observational data for model validation

### 2.5.1 NH$_3$ field measurement flux

A set of 17 days (11$^{th}$ to 27$^{th}$ October 2012) of NH$_3$ flux data from field measurement were used to validate the model. Huo et al.
(2015) conducted a field-scale experiment in the spring of 2012 at a winter wheat cropland, quantifying NH$_3$ emissions from surface fertilization under realistic cultivation conditions. In the field, three types of fertilizers (i.e., urea, ammonium sulfate and compound nitrogen-phosphorous-potassium fertilizer) were used and the fertilization lasted about 20 days for hundreds of divided plots, which have great representation for NCP agricultural situation. The NH$_3$ concentrations were continuously measured at two heights (2.5 m and 8 m) by Picarro (G2103) and Inverse Dispersion Method (IDM) was employed to derive
the heterogeneous NH$_3$ emissions rate. Besides NH$_3$ flux, ancillary environment measurements of air temperature at 2 m, soil

temperature at 0.05 m underground, wind speed and soil water content were also taken. The total input N for urea, nitrogen-phosphorous-potassium and ammonium sulfate are averaged as 140 kg N ha$^{-1}$, 117 kg N ha$^{-1}$ and 122 kg N ha$^{-1}$, respectively. The details of the measurement procedure were described in Huo et al. (2015).

### 2.5.2 Long term NH$_3$ concentration in Beijing and Nanjing sites

Continuous measurements of NH$_3$ and NH$_4^+$ concentration located in Beijing and Nanjing sites of 2019 were used to evaluated the NH$_3$ simulation. In both two sites, the hourly NH$_3$ and NH$_4^+$ were measured by Monitor for Aerosols and Gases in ambient Air (MARGA, MetrohmLtd., Switzerland). In Beijing, the observation is conducted at the Chinese Research Academy of Environmental Sciences (CRAES) (40.05◦ N, 116.42◦ E). In Nanjing, the site is located in the Station for Observing Regional Processes of the Earth System (SORPES) in Nanjing University Xianlin Campus, which is a regional background station in 280 the western part of the YRD region (32.11◦ N, 118.95◦ E) (Ding et al., 2016).

### 2.5.3 Spatial distribution of NH$_3$: IASI and NNDMN

Observations of NH$_3$ spatial distribution from space and ground stations were used in this study. Tropospheric vertical column densities (VCDs) of NH$_3$ were derived from the measurements of Infrared Atmospheric Sounding Interferometer (IASI) on board MetOp-A (Van Damme et al., 2015, 2017; Clarisse et al., 2009). We determined the monthly averages of NH$_3$ column 285 concentrations over the eastern China during 2019, based on the relative error weighting mean method (Van Damme et al., 2014). Surface NH$_3$ concentrations in the NNDMN including 43 observation stations were used to compare with simulation. The land types of the NNDMN sites cover cities, farmland, coastal areas, forests and grasslands. Measurements during the period from January 2010 to December 2015 by the NNDMN were used. Surface NH3 concentrations were measured using an active DELTA (DEnuder for Long-Term Atmospheric sampling) (Flechard et al., 2011).

## 3 Evaluation


### 3.1 Modelling configuration for estimating China's ammonia emission

To evaluate the dynamic NH$_3$ flux model and figure out the aerosol response to dynamic NH$_3$ emission, we designed a pair of parallel experiments by using WRF-Chem. The simulation with the coupled dynamic model is referred to as the "*online*" experiment, while the simulation without the dynamic model is referred to as "*base*" experiment. The base simulation used 295 monthly country-level NH$_3$ inventory based on MEIC NH$_3$ inventory, which is described by Huang et al. (2012). The comparison of online emissions and fix MEIC NH$_3$ inventory map are shown in Figure S2. Other anthropogenic gas emissions from power plant, industrial, residential, and vehicle sectors were taken from the MEIC database (Li et al., 2017). Both two experiments were run for the entire year of 2019 as well as some individual case over the NH$_3$ hot-spot region in eastern China (18° N–50° N, 95° E–131° E) with 20 km grid resolution. For 2019, the running time is from Dec 10$^{th}$ 2018 to Dec 31$^{st}$ 2019, 300 each run covered 24 h and the last hour chemical outputs from the preceding run were used as the initial conditions for the

following run. The first 20 days were regarded as the model spin-up period for atmospheric chemistry, so as to better characterize aerosol distributions and minimize the influences of initial conditions and allow the model to reach a state of statistical equilibrium under the applied forcing (Berge et al., 2001). The initial and boundary conditions of meteorological fields were updated from the 6 h NCEP (National Centers for Environmental Prediction) global final analysis (FNL) data with

a 1°×1° spatial resolution. NCEP Automated Data Processing (ADP) surface and global upper air observational weather data of wind, temperature and moisture are assimilated to better characterize meteorological factor. The setting of each individual cases is also same as above.

The main configurations for the base and online experiments are listed in Table 2. A new version of the rapid radiative transfer model for general circulation model applications (RRTMG) was employed to depict the radiative transfer process for both

shortwave and longwave radiation (Iacono et al., 2008). The Noah land surface scheme (Ek et al., 2003) was used to describe the land–atmosphere interactions, implemented with the Yonsei University PBL scheme (Hong et al., 2006) to describe the diurnal evolution of the planet boundary layer (PBL). As for cloud and precipitation processes, the new Grell–Freitas cumulus ensemble parameterization (Grell and Freitas, 2014) along with Lin microphysics (Lin et al., 1983) accounting for six forms of hydrometers were employed. The WRF simulation was thoroughly evaluated through comparison to comprehensive surface

meteorology data and compiled in a publicly accessible report (U.S. EPA, 2021). As for Chem model, the Model for Simulating Aerosol Interactions and Chemistry (MOSAIC) (Zaveri et al., 2008) and the CBM-Z (carbon bond mechanism) photochemical mechanism  (Zaveri and Peters, 1999) were used. The MOSAIC aerosol scheme includes physical and chemical processes of nucleation, condensation, coagulation, aqueous phase chemistry, and water uptake by aerosols.

**Table 2. WRF-Chem domain setting and configuration selection**

| Domain setting | |
| --- | --- |
| Simulation region | 18°N–50°N, 95°E–131°E |
| Grid spacing | 20 km × 20 km |
| Vertical layers | 29 |
| Map projection | Lambert conformal |
| Configuration selections | |
| Land surface | Noah |
| Boundary layer | YSU |
| Microphysics | Lin et al. |
| Cumulus | Grell-Freitas |
| Radiation | RRTMG |
| Chemistry | CBMZ |
| Aerosol | MOSAIC |

## 3.2 Spatiotemporal pattern of ammonia emission in China

In 2019, the total atmospheric ammonia emission in China was estimated to be 12.67 Tg, and the emission density was around 1.32 Mg km$^{-2}$. The total amount was approximately threefold that obtained for Europe (4.18 Tg) and contributed approximately
38 % of Asian NH$_3$ emissions (Kurokawa and Ohara, 2020). Further, this estimated emission was relatively close to the improved emissions based on AMoN-China and the ensemble Kalman filter (13.1 Tg) (Kong et al., 2019). The most important contributor is livestock manure management (5.25 Tg), accounting for approximately 41.4 % of the total budget. Next is the fertilizer application (5.09 Tg), which was responsible for 40.1 % of emissions. With high nitrogen content of about 46 %, urea fertilizer is the most widely used N fertilizer in China, accounting for 89 % of the total fertilizer. Regarding livestock
waste, the free-range is the largest contributor (65.6 %) to livestock-waste-related NH$_3$ emissions. Next is intensive rearing (approximately 30 %), which refers to the process of raising livestock in confinement at high stocking density, in this process, the farm operates as a factory. Grazing, a relatively less important system without additional feed supplementation, only dominates in the northern and western parts of China.

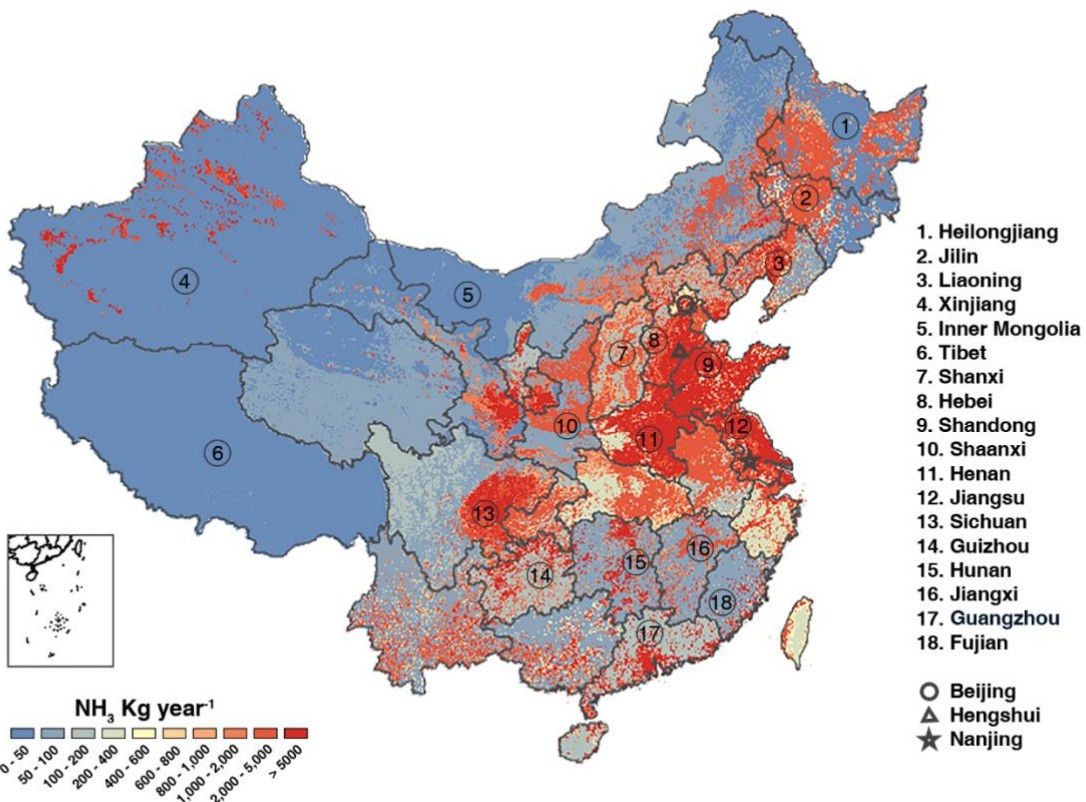

**Figure 2:** Spatial distribution of ammonia emissions with a spatial resolution of 1 km ×1 km grid (kg yr$^{-1}$).

The spatial distribution of $NH_3$ emissions in 2019 is illustrated in Fig. 2. It is clear that the high emission rate concentrates in Shandong, Jiangsu, Henan and Hebei provinces and in eastern Sichuan. Shandong province has the highest $NH_3$ emission density of 4.46 Mg km$^{-2}$, which is 3 times higher than the national average. Multiple-cropping system is a wildly applied agricultural practice in China and varies from region to region. North-eastern China is a mainly single-cropping area, with spring wheat as the main crop. The double-cropping area (~27.74 % of total cropland area) is mainly distributed in the North China Plain (NCP), with a 25,624,000 and 5,754,000 ha area under cultivation of cereal crops and vegetables, respectively, is responsible for 49 % of the $NH_3$ emissions from fertilization in China. Triple-cropping and limited triple-cropping systems are prevalent in southern China due to the tropical subtropical monsoon climate with high temperature and ample rainfall. With intensive cultivation, the cropping land in Guangdong, Fujian, Hunan and Jiangxi provinces also shows large $NH_3$ volatilization. Paddy fields are common in this region, with early rice and late rice being the dominant crop and contributing the most emissions. The smaller emitters are located mostly in western China, with a minimal amount of arable land or low use of synthetic nitrogenous fertilizers.

The spatial distribution of $NH_3$ emissions from livestock waste is similar to that from synthetic fertilizers, with high emission rates in NCP, eastern Sichuan, and western parts of Xinjiang province. Besides intensive agriculture fertilization, Henan, Shandong, Hebei and eastern Sichuan provinces are well-known for their large animal population. In Henan, Hebei and Shandong, many kinds of animals are extensively bred like beef, dairy, pork and poultry, providing approximately 0.38 Tg, 0.31 Tg and 0.48 Tg ammonia emissions, respectively. Sichuan province is also a large emitter (0.30 Tg), with cattle and pig as the main animals accounting for most emissions. In Xinjiang provinces, sheep are widely raised, which is responsible for remarkable ammonia emissions related to sheep manure management. Cattle are widely raised in northeast China and are responsible for around 50 % of the $NH_3$ emission in Liaoning and Heilongjiang provinces.

The peak of $NH_3$ emission over NCP might be the joint result of intensive agricultural activities and environmental conditions like high soil pH and wind speed and less rainfall. In China, the soils in the NCP are mainly neutral (pH 6.5−7.5) or alkaline (pH >7.5), and soils distributed in southern or north-eastern China are mainly acid soil (pH 5.5−6.5) or strong acid soil (pH < 5.5). $NH_3$ volatilization increases significantly with an increase in soil pH, due to the high potential for $NH_3$ emission (Ryan et al., 1981). Thus, the relatively high pH in northern China is another key reason for the large flux in the NCP. Besides soil acidity, $NH_3$ volatilization increases exponentially with wind speed. Spatially, northern China features frequent strong wind. The annual mean wind speed varied from 4.64 m s$^{-1}$ north-eastern China to 3.55 m s$^{-1}$ southern China (Liu et al., 2019). Higher wind speed leads to rapid $NH_3$ release from soil to the atmosphere, which may contribute to the high ammonia emissions in the northern part of China. The annual mean meteorology-dependant dynamic CFs over eastern China are shown in Figure S2(a), for both northern China and northwest China the CFs are greater than 1, indicating that meteorological conditions in north are more conducive to ammonia emissions.

In addition to great regional disparities, China's $NH_3$ emission is also characterized by obvious seasonality. The monthly variation in Fig. 3 clearly indicates that the emissions were primarily concentrated from April to September due to the intensive agricultural activities and high temperature. In China, the new spring seeding generally begins in April. During this period,

spring wheat, soybeans and cotton are sown in the single-cropping area with a large amount of N fertilizer applied to the cropland as the base fertilizer. In the following 1–2 months, due to the application of top fertilizer and warming air temperature, $NH_3$ emissions tend to continuously increase to August. From summer on, the winter wheat–summer maize rotation system begins in the NCP. The winter wheat–summer maize rotation system has been practiced as a characteristic farming practice,

producing about 60 % of the total wheat and 33 % of the total maize production in China (Zheng et al., 2021). There are two cropping cycles for the NCP region beginning in summer and late-autumn, respectively. June to August are one of the main cropping periods in the NCP, with the sowing, basal dressing and top dressing of summer maize. The seeding and basal dressing of winter wheat begin in mid-September and the top dressing is applied 2 months later, which could be responsible for the high emission rate during the autumn season. Meanwhile, in southern China with a large triple-cropping area, fertilization of

early and late rice was carried out intermittently from April to mid-October. Most of the crops begin to harvest in the autumn, which lead to a declining emission since then. Different from the hot seasons, because of the less $NH_3$ volatilization related to lower temperatures and relatively rare cultivation during the winter, the $NH_3$ emissions decreased to one-third of summer levels.

Figure 3 also shows that the monthly national average $NH_3$ emission in China features obvious day-to-day variation with large

disparities between 10th and 90th percentiles, especially during hot seasons. Such a large discrepancy in daily emission rate could be attributed to the highly fluctuating daily weather condition and the associated meteorological parameters throughout the year, and the existence of the summer monsoon makes China have more obvious daily emission changes in summer. However, the fixed $NH_3$ emission inventories that are wildly utilized currently assume no daily variation of emission rate, and thus could not represent the day-to-day variation of emission rate under real weather conditions.

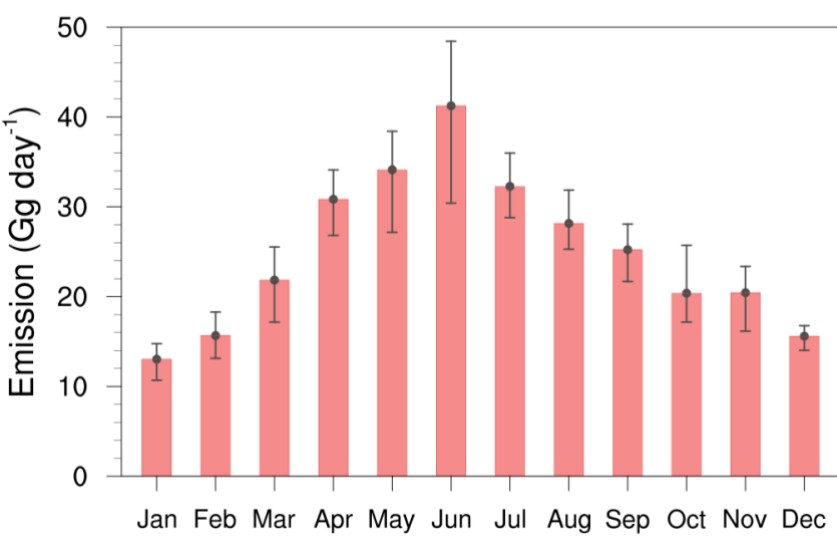


**Figure 3:** Seasonal pattern of averaged daily ammonia emission in eastern China (18°N–50°N, 95°E–131°E). The lower and upper points of vertical lines show the 10th and 90th percentiles, respectively.

**3.3 Comparison of hourly emission flux with field experiments**

The most important advance of this model is the online calculation of highly time-resolved emission rate, available NH$_3$ emission flux at high temporal resolution was also collected to further validate the model. Air-surface exchanges of NH$_3$ were measured over a fertilized wheat canopy in the spring of 2012 in Hengshui, China. In the field, different plots of the cropland were cultivated by different farmers and the fertilization practices were conducted plot by plot. Ammonia flux measured from this kind of realistic cultivation conditions can represent real agriculture emission in the NCP.  The high temporal resolution

variations of measured NH$_3$ flux and simulated flux are shown in Fig. 4. The mean observed and modelled fluxes during this period were 0.38 kg km$^{-2}$ hr$^{-1}$ and 0.63 kg km$^{-2}$ hr$^{-1}$, respectively. In Fig. 4(a), during 16th–17th October, the flux significantly decreased during the experiment for the duration of the rain. A similar tendency of low NH$_3$ flux during rainy days was found during observations, as reported from other studies (Osada, 2020; Roelle and Aneja, 2002). Increased contents of soil pore water dilute NH$_4^+$ in the liquid phase and inhibit its evaporation as NH$_3$. Furthermore, wet surfaces of cuticular leaves absorb

ambient NH$_3$ under high relative humidity during rain. The model was able to reproduce the low emission during the rain. In Fig. 4(b), the online dynamic model agrees well with diurnal field flux for the median values, while the 95th percentiles are much lower during daytime. The model captured the observed diurnal trends well with emission dominating during the day generally peaking at around the time of the maximum daily air temperature (12:00–14:00 local time). However, the model seemed to overestimate the measurements during the late morning when the measurements were dominated by deposition.

Since the model only estimates unidirectional emission flux, the major deposition flux during the 9:00–10:00 local time is not included in the model results, which leads to the overestimation of net emission flux by the model.

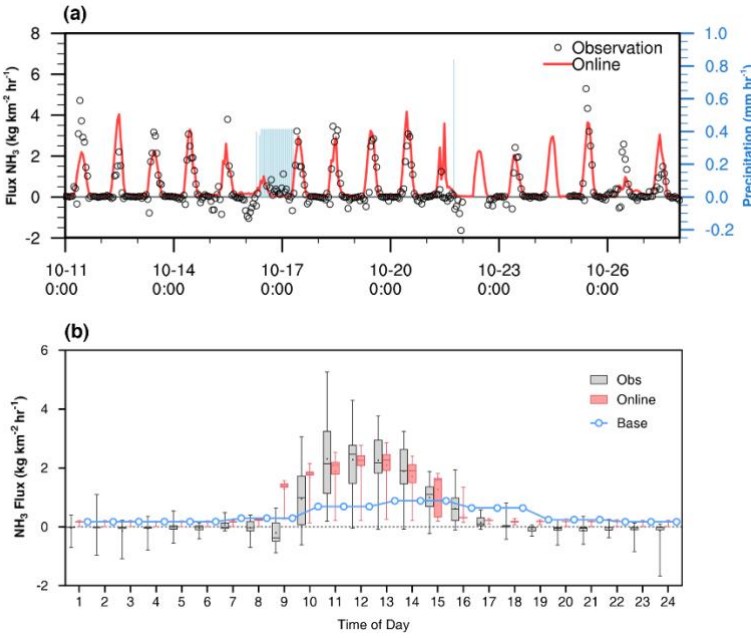

**Figure 4:** (a) Time series of observed (black symbol) and WRF-Chem Online model NH$_3$ flux (red line) above China Hengshui agri-field from the 11$^{th}$ to 27$^{th}$ October 2012. (b) Diel hourly box plots of observations flux measurements (grey), paired with online model results (pink) and base model results (blue). The 5$^{th}$ and 95$^{th}$ percentiles are represented by the whiskers, the 25$^{th}$ and 75$^{th}$ quantiles are enclosed in the box, the median is represented by the horizontal line through the box, and mean value is the dot in the box. Diurnal profile of emissions from agriculture is applied in base experiment following Du et al. (2020).

Due to weather change during the experiment, the diurnal NH$_3$ emission varied considerably with the large disparities between 5$^{th}$ and 95$^{th}$ percentiles, although at the same time of day. Despite the underestimation of the 5$^{th}$ percentiles, online emission results have obvious distribution with at least 25 % overlap with observation for each hour. However, fixed inventory used in the base simulation are monthly and has no diurnal variation of emission. To integrate this inventory into WRF-Chem simulation, we adopted a diurnal profile with 80 % of NH$_3$ emissions in the daytime, following previous studies (Du et al., 2020). Obviously, the single line diurnal variation of the base experiment is underestimated and could not represent ammonia emission under real weather conditions.

### 3.4 Monthly variation of ammonia concentration and validation by in-situ observations

This newly-developed model in this work is capable of simulating varying NH$_3$ emission rate with changing climate, and monthly variations in the emission rate are compared with available observations. Figure 5 shows the monthly average of ground-site NH$_3$ with model results extracted at Nanjing and Beijing site locations. The observed NH$_3$ variations at two sites clearly demonstrated a minimum in winter and a maximum in summer that reflects the enhanced emission in growing season with higher temperature and denser fertilization. In the Nanjing site, the concentration of observed NH$_3$ ranged from 2.2 ppb in December to 23.0 ppb in July, with the annual average of 11.8 ppb. The small peak in April may be partly explained by the local seeding and fertilization of early rice. However, for base simulation, the NH$_3$ range was 0.02−14.26 ppb and the annual average was 4.99 ppb, which greatly underestimates NH$_3$ concentrations particularly for the warm season (e.g., May−September) (Fig. 5a). The base simulation NH$_3$ increases slightly from March to May, unable to reproduce the peak in April at all. Compared to the base run, the model with online flux shapes better seasonal variation, and the April-July averaged concentration has a similar magnitude as the observed NH$_3$ values (Fig. 5b). The interquartile ranges (shed width) are closer to the observations, and the proportion of overlaid area between observed and online shadows is reaching about 50 %, which is significantly larger than the base. Besides, the online experiment can capture the small peak in April with similar magnitude. For the Beijing site, the annual level of NH$_3$ is 14.2 ppb, which is higher than that in Nanjing site. This is consistent with the higher emission in the northern region than the southern region. The observed NH$_3$ concentration exhibits a large increase from 0.4 in January to 38.3 ppb in June. In contrast, the base simulated monthly ammonia concentration almost flat throughout the whole year, which could not reproduce the observed significant seasonal variation at all (Fig. 5c). By comparison, online simulations notably narrowed the gap with observations, especially from March to September (Fig. 5d). The mean bias of NH$_3$ simulation was 10.5 and 8.2 ppb for the base and online experiments, respectively. However, for the winter seasons, from

October to January, the online experiment overpredicts NH$_3$ at the Beijing site by an average of 95 %. This overestimation may be due to the meteorological or chemical simulation bias from WRF-Chem (Du et al., 2020; Liu et al., 2021) since both online and base experiments show overestimations of the Beijing site in winter.

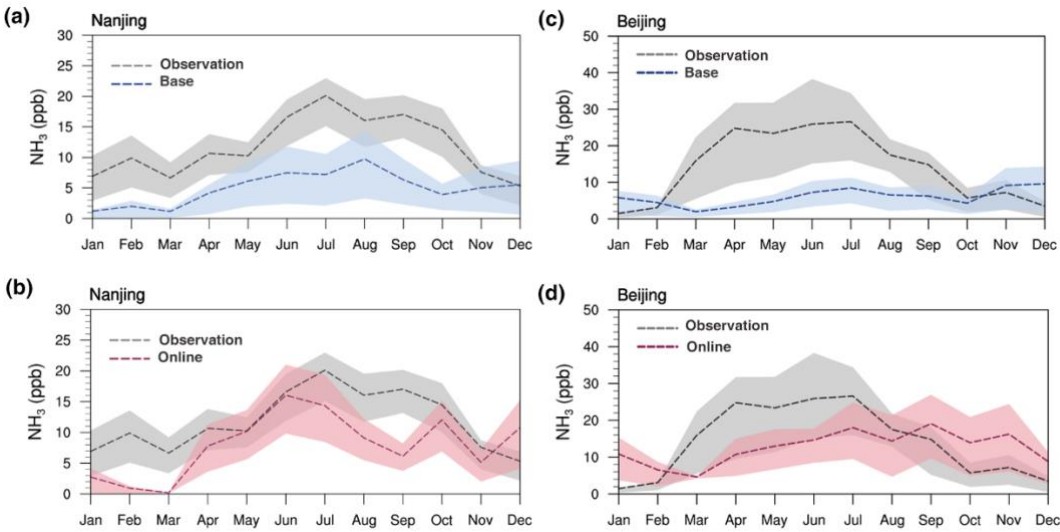

**Figure 5:** Monthly average ammonia concentrations at 2019 Nanjing site for base (a) and online (b) model runs and measurements from local. The dash lines indicate mean values. Lower and upper ends of the filled area indicate the 25$^{th}$ and 75$^{th}$ percentiles of each distribution. (c) and (d) are in Beijing site.

### 3.5 Evaluation of spatiotemporal pattern by monitoring network and satellite retrievals

To better validate the model performance on the spatial and temporal pattern of NH$_3$ emission, we then collected NH$_3$
concentration measurements across China and compared them with the corresponding result from WRF-SoilN-Chem model. Ideally, the model results and observed data should be in the same period in 2019. However, NH$_3$ is not routinely measured at national networks, so there is a lack of published observations for the last five years in China. Since the main emission source of atmospheric NH$_3$ and the activity level would not vary a lot in a short time, and the meteorological parameters for 2019 were closed to mean state of the multiple-year average for 2010s (Fig. S3), therefore, a database of atmospheric nitrogen
concentration from the nationwide monitoring network (NNDMN) between 2010 to 2015 is used to evaluate the spatial pattern and magnitude of surface NH$_3$ concentrations in China (Xu et al., 2019). Figure 6 shows spatial plots of annual observed NH$_3$ by NNDMN and two sets of model results. In Fig. 6(a), the base experiment underestimated NH$_3$ concentration in areas with a high emission density. For instance, the NH$_3$ concentrations in sites located in the southern region of Hebei province and the north of Shandong province were underestimated by ~15 ppb. Besides, the base experiment cannot reproduce the high level
of NH$_3$ in the Pearl River Delta region (PRD). Nonetheless, in southern China, the base experiments overestimated NH$_3$ concentration by ~5 ppb for the two northern sites in Hubei province. In Fig. 6(b), the spatial accuracy of online experiment is

better than that of the base fixed inventory. The online model well reproduces the overall pattern of high values in the NCP, the Sichuan Basin and PRD and low values in other regions.

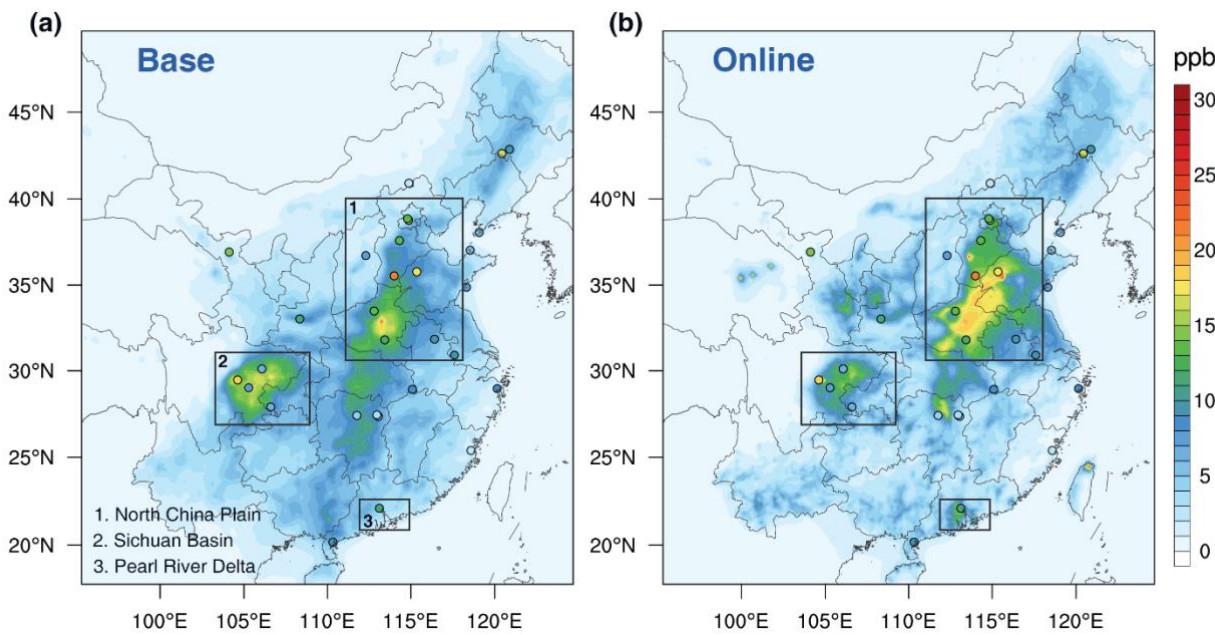


**Figure 6:** Spatial patterns of annual NH₃ concentrations over eastern China. The circles represent measured NH₃ concentration at Nationwide Nitrogen Deposition Monitoring Network (NNDMN) averaged for 2010-2015. The contour plots represent model NH₃ in 2019 from the base experiment (a) and online experiment (b) of this study. The black rectangles represent the three focused regions in eastern China.

Additionally, the Infrared Atmospheric Sounding Interferometer (IASI-A), launched aboard the European Space Agency's MetOpA in 2006, has observed atmospheric NH₃ at a global distribution and bi-daily resolution (09:30 and 21:30), here we use morning observations when the thermal contrast is more favourable for retrievals (Van Damme et al., 2014). The monthly NH₃ vertical column densities (VCDs) were determined based on the relative error weighting mean method (Van Damme et al., 2014). The NH₃ VCDs from the simulations were calculated by integrating NH₃ molecular concentrations from the surface

level to the top of the troposphere at 09:00 and 10:00 local time. As shown in Fig. 7, the online WRF-SoilN-Chem model was able to capture the general spatial distribution of NH₃ VCDs, including the higher concentrations over eastern China relative to western China, as well as the hot-spots over the NCP and the Sichuan Basin. As for temporal variation, monthly average NH₃ concentrations from online run compared to the IASI data showed similar seasonal cycles with highest concentrations in summer, which is consistent with the time of agriculture activities and high ambient temperature. However, the online

experiment slightly underestimated the NH₃ concentrations in spring (April) and winter (December) and a similar phenomenon was found by Li et al. (2021) and Liu et al. (2018). The difference between our model results and satellite observed distributions could be attributed to the uncertainty of NH₃ emission model and biases of satellite inversion algorithm. In satellite VCDs

algorithm, the relative error weighting mean method always biases a high result due to the smaller relative error in a larger column (Van Damme et al., 2014).

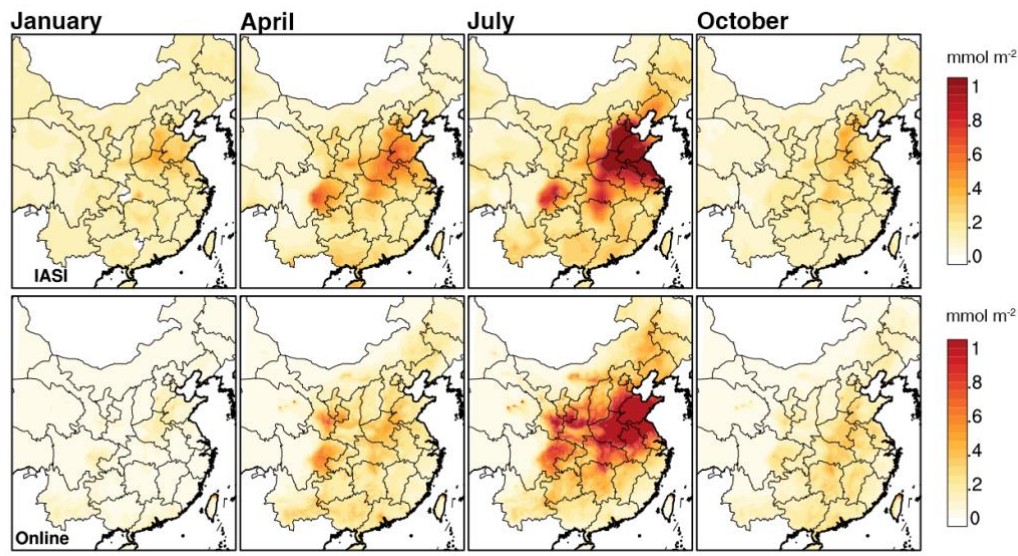


**Figure 7:** Comparison of column NH$_3$ concentrations between IASI satellite retrievals (upper panel), and WRF-Chem online (lower panel), averaged for January, April, July and October 2019.

### 3.6 Improved simulation of ammonia and secondary inorganic aerosol on synoptic scale

Once the gas ammonia is released into the atmosphere, some of them could actively participate in the atmospheric chemical
reactions. As aforementioned, NH$_3$ is a very important alkaline constituent in the atmosphere and is a key precursor to the neutralization of gaseous sulfuric acid and gaseous nitric acid in the atmosphere to form the secondary inorganic aerosols, like sulfate, nitrate and ammonium, which contribute to ambient particulate matter. The amount that becomes aerosol ammonium (NH$_4^+$) depends on the concentrations of anions in the air, typically sulfate, nitrate, and chloride, which can form (NH$_4$)$_2$SO$_4$, NH$_4$HSO$_4$, NH$_4$NO$_3$, and ammonium chloride (NH$_4$Cl). Thus, an evaluation of ammonia flux would include both gas and
aerosol forms (NH$_x$ = NH$_3$+NH$_4^+$). To investigate the response of NH$_3$ emission and aerosols to significant synoptic change, we choose two pronounced cases to evaluate the performance of the modified online model.

Figure 8(a) shows a typical case of NH$_x$ episodes from 1$^{st}$ April to 8$^{th}$ April 2019 in Nanjing. There is a close association between the dramatic air temperature change and NH$_x$ surface level. To be specific, NH$_x$ concentration increased from 15 µg m$^{-3}$ at air temperature below 10 °C to more than 30 µg m$^{-3}$ at temperatures higher than 20 °C, suggesting a promotion effect
of higher temperature to NH$_3$ emission and NH$_x$ concentration in surface. In online emission, the NH$_3$ emission flux also shows dynamic variation and the pattern is quite similar to the temperature and NH$_3$ concentration (Fig. 8b). Similar phenomena of a strong relationship between temperature and ammonia emission has also been proved by previous laboratory experiments (Clay et al., 1990; Pedersen et al., 2021; Niraula et al., 2018). However, since the resolution of the offline inventory was only

monthly, the emission intensity showed only monthly differences between March and April, which could not reproduce the
variation of daily dramatic emission rate at all. Overall, the model with meteorology-dependent mechanism well tracked the
NH$_x$ variation trends in observation period, and remarkably improved NH$_x$ simulations with the MB decreasing from –9.5 to
0.7 μg m$^{-3}$ in NH$_x$ concentrations. Figure 8(c) shows the soil moisture and measured NH$_x$ concentration variation from 1$^{st}$ to
18$^{th}$ January 2015 in Beijing site. In this case, the local NH$_x$ concentration and soil moisture have similar variations with
reaching peak together on 8$^{th}$ January and 15$^{th}$ January. The variation of online ammonia emission rate also has similar pattern
to soil moisture (Fig. 8d). The base simulation without considering soil moisture's effect significantly underestimates the NH$_x$
peak concentration by ~30 μg m$^{-3}$. The online data can greatly capture the magnitude and temporal variation of NH$_x$
concentration with the MB decreasing from –12.4 to –3.1 μg m$^{-3}$. As the precursor gas, the performance of NH$_3$ simulation
directly affects the formation of simulated secondary inorganic aerosols like NO$_3^-$. In Nanjing and Beijing, the base experiment
shows nitrate aerosol was generally underestimated with MB of –5 μg m$^{-3}$ due to the lack of NH$_x$ (Fig. 9). The online model
generally reproduced the observed nitrate concentration, with a small mean bias from –4.5 to 1.9 μg m$^{-3}$ in Nanjing case and
from –5.2 to 0.8 μg m$^{-3}$ in Beijing case. Through the diagnostic analysis of WRF-Chem, the chemical reaction between NH$_3$
and HNO$_3$ was the main reason the nitrate pollution in both cases (Figure S4). That is to say, nitrate formation in this region
is highly sensitive to the ambient NH$_3$ availability. In both cases, the online emission rates were higher than base emission and
the nitrate and total ammonia better simulated, indicating that the traditional emission inventories may be underestimated.
Comparison between fixed emission input and online emission modelling clearly demonstrates that the numerical description
of highly time-resolved NH$_3$ emission has a superior performance on the magnitude and temporal variation of secondary
aerosol.

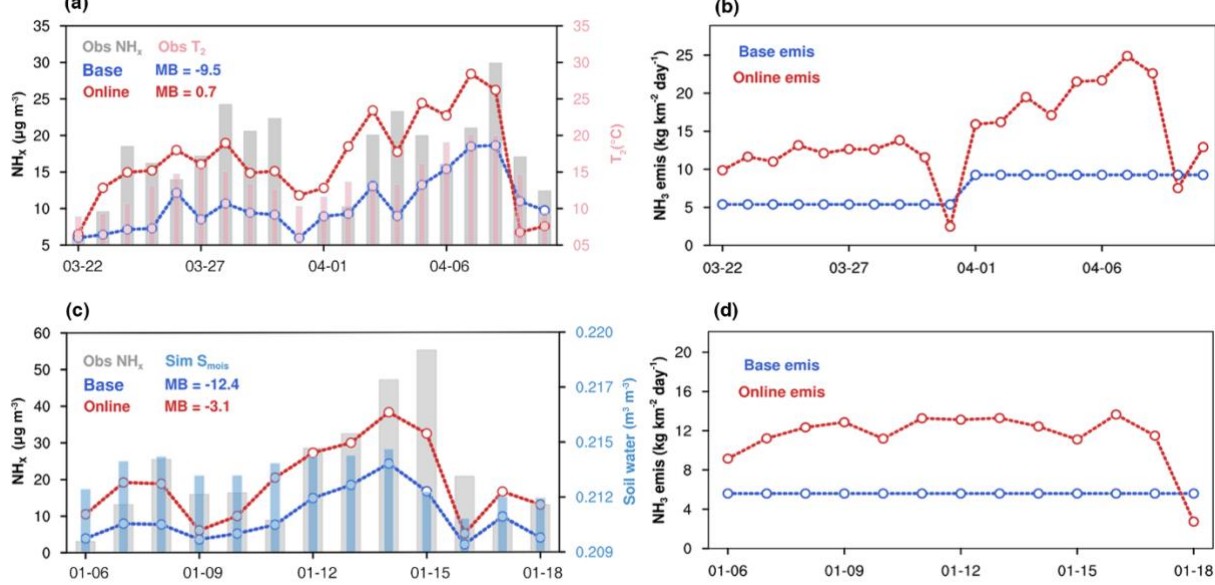

**Figure 8:** (a) Time series of total ammonia (NH$_x$) concentrations from 22$^{nd}$ March to 11$^{th}$ April 2019 in Nanjing site. The T$_2$ represents the air temperature at 2m ground level from local site. (b) Daily NH$_3$ emission rate same as (a) in Nanjing site. (c) Time series of NH$_x$ concentrations from 6$^{th}$ to 18$^{th}$ January 2015 in Beijing site. S$_{mois}$ represents the top 5cm thick layer soil water which is derived from WRF model calculations. The mean bias of modelled concentrations is labelled by MB. (d) Daily NH$_3$ emissions rate same as (c) in Beijing site. Blue lines represent the MEIC NH$_3$ emission used in base simulation, red lines represent online emissions from WRF-SoilN-Chem.

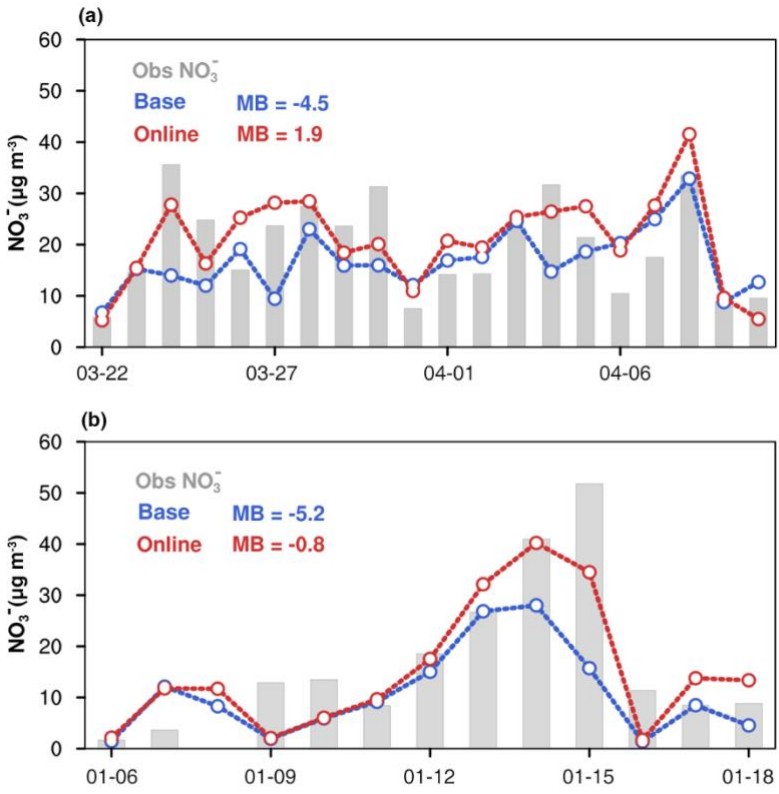

**Figure 9:** (a) Time series of nitrate (NO$_3^-$) concentrations from 22$^{nd}$ March to 11$^{th}$ April 2019 in Nanjing site. (b) Time series of nitrate (NO$_3^-$) concentrations from 6$^{th}$ – 18$^{th}$ January 2015 in Beijing site.

## 4 Summary and Future work

We developed the WRF-SoilN-Chem (v1.0) model, which is an online coupling of the WRF meteorological model and dynamic NH$_3$ flux model, to simulate meteorology-dependent regional NH$_3$ emission at high spatiotemporal resolution. In this model, high resolution basic emission data and meteorology-dependant parameterizations were implemented, and soil temperature, wind speed, soil moisture and rainfall simulated from WRF were considered as the important factor in the

parametrization to control $NH_3$ emission. This version can be easily implemented in other regional or global models and could
serve as a tool for more precise and highly time-resolved estimation of the $NH_3$ emission calculations.

The preliminary evaluation by multiple ground-based and satellite observations indicates that the WRF-SoilN-Chem model is able to better represent the spatiotemporal variation of surface $NH_3$ concentrations over eastern China than the widely used offline MEIC $NH_3$ inventory. We evaluated the model by field experimental data, monitor observation and satellite retrievals. Compared to the monthly offline inventory, this simple model well reproduced the diurnal variation of $NH_3$ flux measured in
the typical farmland. Our first application showed that the WRF-SoilN-Chem model was able to capture the magnitude spatiotemporal variation of $NH_3$ surface and column concentrations over China in 2019. And this model has smaller biases in the simulation of ammonia at both Beijing and Nanjing monitoring stations. The seasonal variations from the online calculation in WRF-SoilN-Chem model are distinct and have similar patterns as observations which are characterized by high concentrations in summer and low concentrations in winter.

The consideration of meteorological factors makes the model more accurate for ammonia simulations under conditions of drastic weather changes. To be specific, the mean bias of $NH_x$ simulation during Nanjing and Beijing case period were $-9.5$ and $-12.4$ $\mu g\ m^{-3}$ without the influence of dynamic meteorological factor, reducing to $0.7$ and $-3.1$ $\mu g\ m^{-3}$ when considering temperature and soil moisture's effect. Besides, higher and precise $NH_x$ concentration favors the formation of nitrate by enhancing gas-particle conversion, reducing the mean bias of nitrate concentration from $-4.5$ to $1.9$ and from $-5.2$ to $-0.8$ $\mu g$
$m^{-3}$ in Nanjing and Beijing, respectively. In general, with the met-dependence mechanism, the online model can optimize the simulation of surface $NH_x$ and $NO_3^-$ under dynamically changing weather.

Despite providing more accurate and high-resolution estimation of $NH_3$ emission, the current version of the WRF-SoilN-Chem (version 1.0) still has some limitations including (1) the basic EFs were assumed to be the same throughout the month. However, in reality, the soil pH and nitrogen content of the soil after fertilizer application usually increases rapidly under the hydrolysis
of urea and gradually depletes, which leads to variation in EFs as well. So, the constant basic EFs could underestimate the peak emission after fertilization. (2) the meteorological CF parameterization scheme used in the model same for all agricultural soil. However, the emissions can be different from soils with the same water content but different porosity (soil water content at saturation), which is not considered in the model. (3) the gradual decay of $NH_3$ emissions after fertilization is not added in the model, because the specific fertilizer application date for each agriculture plot is not accessible. The model can be updated
and further developed as more laboratory or field measurements data are accessible. The planned further improvements include (1) the option to use bidirectional $NH_3$ flux model based on a resistance approach; (2) update of the dynamic emission factor parameterization with more field experiment results; (3) development of soil emission modules for other atmospherically reactive species, such as HONO and $NO_x$. Such improvements could make the WRF-SoilN-Chem an increasingly useful tool to analyse $NH_3$ emission and its impact on air pollution and biogeochemical nitrogen cycling.

## Code and data availability

NH$_3$ vertical column density data are freely available through the AERIS database: https://iasi.aeris-data.fr/NH3/. The Nationwide Nitrogen Deposition Monitoring Network data are publicly available at https://doi.org/10.1038/s41597-019-0061-2. The standard WRF-Chem v3.9 is freely accessible to the public by following https://github.com/wrf-model/WRF/releases. The source codes of dynamic NH$_3$ emission model and input files about basic emission file that are presented in the article have been archived and made publicly available for downloading from https://doi.org/10.5281/zenodo.7134286.

## Author contributions

XH, YS and TZ conceived the idea and guided the study. CR and XH did model construction, simulations, and paper writing. YS provided the field experiment and flux measurement. XL and ZW provided NNDMN monitoring data of ammonia concentration. All authors contributed to the writing and editing of this paper.

## Competing interests

The authors declare that they have no conflict of interest.

## Financial support

This work was supported by National Natural Science Foundation of China (41875150), the Ministry of Science and Technology of the People's Republic of China (2022YFC3701105), and the Fundamental Research Funds for the Central Universities (14380187).

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
