# Peer review of "A dynamic ammonia emission model and the online coupling with WRF-Chem (WRF-SoilN-Chem v1.0): development and regional evaluation in China"

_Geoscientific Model Development, 2022_

## Referee Comment (RC2)

The manuscript "*A dynamic ammonia emission model and the online coupling with WRF-Chem (WRF-SoilN-Chem v1.0): development and evaluation*" by Ren et al. describes an ammonia emission model coupled with WRF-Chem that incorporates impacts of meteorological and soil conditions on emission factors (EFs), which aims to provide better estimates of NH3 emissions on both spatial and temporal scales. The modelling results (including fluxes and concentrations) were compared to multiple observations and measurements in China, with evident improvements that can be seen. This study addresses the lack of consideration of environmental impacts on NH3 emissions in current emission inventories. The main assets of this study include: 1) Very simple correction factors were used to represent the climatic dependences of NH3 emissions but covered important processes. 2) The emission model was performed on high spatial and temporal resolution and was coupled to an online climate-chemistry model, which is an advantage for predicting NH3, a short-lived species whose sources can be greatly influenced by meteorological factors. 3) Improvements in estimating NH3 emissions and simulating atmospheric chemistry were made by implementing this emission model. 4) The operation of coupling the emission model with the parent WRF-Chem model looks user-friendly according to the description in the manuscript. Overall, the manuscript is clearly structured and well written. The manuscript should be published in GMD after the authors address the following questions and comments.

**Major comments**

1. As mentioned, the emission model is simple but it accounts for important factors that influence $NH_3$ emissions. Therefore, it is crucial to justify why to use the parameterizations shown in the manuscript. The model focuses on China geographically, and most of the corrections for EFs are empirical equations. To what extent these parameterizations can be used globally or in other regions to give reasonable estimates remains unclear. Is it only applicable to China? How well is the model performance in other regions/countries if applying the correction factors in the same way? A useful method would be performing sensitivity tests of some selective parameters in the equations to justify which are the most important ones impacting emissions and what is the relative importance of each parameter. The method used in this study is modifying existing EFs from emission inventories by including a set of environmental dependencies. Theoretically, this can be done for regions like Europe and US, which could be a future direction.

2. The description of running the emission model and the WRF-Chem model in the manuscript is unclear. Did you average the monthly (or annually) basic emission data to obtain the emission with the temporal resolution that is required in the WRF-Chem model? In addition, how did you run the coupled model for China? I assume that you ran nested simulations (I could be wrong). If so, what were the boundary conditions and what emissions did you use for the global run? It is useful to add a section or a paragraph to clarify.

3. Following the above points, a potential weakness I am concerned about is the overall integrity of the emission model. The emission is not calculated from the sources such as the amount of nitrogen in the fertilizers or manure but is derived from the given EFs. However, the EFs are related to the sources such as how much fertilizer nitrogen is applied on land. If you calculated the online EFs at each time step (incorporating the correction factors into the basic emission), the hidden philosophy of the model is that there is a source at every time step which is problematic. Meanwhile, the model might overestimate the emission because the nitrogen reservoirs are depleting. If the basic EFs were assumed to be the same throughout a period (e.g., a month), this does not reflect the tendency of a decreasing emission as there is less and less nitrogen to be emitted. This should be discussed in the manuscript either as an uncertainty evaluation or how the model deals with such a problem.

4. There are some inconsistencies and ambiguities in the overall design of the experimental simulations. It is important to clarify in the manuscript why the year 2019 was chosen to run simulations (rather than a year between 2010 – 2015) and for each comparison what meteorology was used. In the evaluation section, both the annual $NH_3$ concentrations over eastern China by NNDMN (Fig.4) and the site measurements (Fig.8b and Fig.9b) used for model comparisons were not from the year 2019. This raises the question of whether the comparisons are representative enough when using different meteorology for the simulations given that $NH_3$ is strongly influenced by climatic conditions.

It is stated in Section 3.1 that "…experiments were run for the entire year of 2019 …", but it is not clear which year's meteorology was used for the comparisons with satellite (Section 3.3; Fig.5) and field measurements of $NH_3$ fluxes (Section 3.5; Fig.7). Also in Section 3.5, for comparing to the field measurements of $NH_3$ flux, did you use the corresponding meteorological in 2012?

5. In Section 3.2, it is unclear whether the description/discussion is for basic $NH_3$ emissions or online $NH_3$ emissions. It is useful to include a map for the online $NH_3$ emissions in Figure 2. Besides, it can be helpful to see the difference between the two.

6. In Section 3.4, since the model results can be extracted at site locals, such as Nanjing and Beijing, can you present more comparisons in the same way for other sites (as there are over 10 more monitoring sites shown in the boxes in Figure 2)? This can be put into Supplementary materials. Or it is worth adding a paragraph or a few sentences to clarify why only these two sites were selected for comparisons, i.e., availability, data quality etc.

7. In Section 3.5, I was wondering how you ran the model to get the emissions. First, as mentioned, which year's meteorology was used? Second, did you run the model at the site scale or did you just extract results from the regional simulations for China? How you calculated the basic EFs from the nitrogen application rates given by the field study, i.e, $EF_{0i}$, $CF_{pH,}$ $CF_{method}$, and $CF_{rate}$ in Equation 2 are not well described. Details can be put in Supplementary Materials. Third, the field measurement of $NH_3$ emissions shows relatively comparable magnitudes in terms of daily emissions, i.e., the general daily trend of $NH_3$ emissions is quite "flat" rather than gradually decreasing over 15 days. The field emissions after fertilization usually reach the maximum within a few days and then decline due to less nitrogen being available for emitting. It is unknown if the model is capable of capturing such features. In other words, the impressive agreement between the modelled emissions and measurements becomes a bit less convincing. It might be due to averaging the monthly emission factor giving the same hourly emission factors, which results in comparable daily sums. Nevertheless, the model reproduces diurnal variations in emissions very well and captures the decreasing feature of $NH_3$ during a rainy day.

8. There is a lack of discussion on uncertainty from the new emission model. This can be either some text descriptions discussing various uncertainties, or any back-of-envelope calculations derived from the potential sensitivity tests.

9. Although WRF-Chem can be run globally, the study focuses on China. I would suggest revising the title of the manuscript to be more specific on its spatial coverage. For example: "*A dynamic ammonia emission model and the online coupling with WRF-Chem (WRF-SoilN-Chem v1.0): development and regional evaluation in China*" or "*A dynamic ammonia emission model for China and the online coupling with WRF-Chem (WRF-SoilN-Chem v1.0): development and evaluation*".

10. I would personally suggest putting Section 3.5 in front of Section 3.3. Section 3.5 directly compared modelled $NH_3$ emissions to measured emissions, while Sections 3.3 and 3.4 are comparisons between modelled and observed/measured atmospheric $NH_3$ concentrations. By doing that, the evaluation is in the order of "emission, $NH_3$ concentrations, and aerosol concentrations". This point is optional.

**Specific comments**

P6L138: This is not entirely correct because soil pH can increase after urea application. Urea hydrolysis consumes $H^+$ ions so it tends to lead to more $NH_3$ emissions. The extent of pH increase is dependent on the soil's pH buffering capacity. Although this is complex, it should be specifically clarified that the model assumes constant soil pH. Please modify "In the fertilizer application section, soil pH … are relatively stable in a short time."

P6L164-165: Same comments as mentioned above and in the 3rd points from Major comments. Ammoniacal N concentrations can vary greatly throughout the application period. Ammoniacal N is lost from the soils through various pathways such as volatilization, nitrification, and physical transport like runoff and diffusions, which can affect the concentration of ammoniacal N. Ammoniacal N also exists in different phases, e.g., can be adsorbed on soil particles, which depends upon soil moisture and other factors. Since the model uses simple correction factors and does not include detailed soil processes, it should clearly state the assumptions to avoid misleading. Please consider rephrasing into "…ammoniacal N concentration and soil pH **are assumed to be** stable in short time …".

P7L177: Is there any statistical or activity data for China supporting that "the handle of excrement is usually settled in closed containers"?

P7L192: Please explain why soil moisture correction is applied for modelling emissions from animal houses. There are houses with concrete floors or slatted floors. Or is this equation only applied to specific management?

P8L211-216: It is useful to include more details for the experimental study carried out by your research group which is used for deriving soil temperature correction factor, i.e., what method was used? How was the study designed? Any reference?

P8L225-229: It is worth mentioning in the manuscript that emissions can be different from soils with the same water content but different porosity (soil water content at saturation), which is not considered in the model.

P9L230-235: I was wondering what is the underlying mechanism that rainfall affects $NH_3$ emissions in the model. Is it because of infiltration, runoff, or changes in soil moisture? Is there a double-counting issue here between the rainfall and soil moisture correction?

P11L268: Again, since urea is the most widely used fertilizer in China, a discussion of the uncertainty caused by not including soil pH change is missing.

P13L293-294: Tibet has very little $NH_3$ emissions as shown in Figure 2, while there are some hot spots in Xinjiang province. Meanwhile, sheep are not a significant contributor to $NH_3$ emissions especially when grazing is dominant. What does the model tell? You should be able to diagnose the sectoral emissions from the model.

P14L337: Then why not simulate 2010-2015?

P19L431-432: As stated in the manuscript, there are no diurnal variations in the inventory. However, some diurnal variation can be seen as shown in Figure 7b from the base run (e.g., 10:00 to 18:00 has higher emission than other times), which is confusing.

**Other comments**

P2L63: "environment elements" to "environmental elements".

P5L124: "agriculture soil" to "agricultural soil".

P9L235: Numbering of the equation is missing.

P18: Labelling of Figure 7 is missing.

P18: Consider using grams or kilograms rather than mol.

---

## Author Comment (AC2)

[Figure]

Figure R1: (a) Time series of total ammonia (NH₃) emissions from 22ⁿᵈ March to 11ᵗʰ April 2019 in Nanjing site. (b) Time series of total ammonia (NH₃) emissions from 6ᵗʰ to 18ᵗʰ January 2015 in Beijing site. Blue lines represent the MEIC inventory used in Base simulation, red lines represent Online emissions from WRF-SoilN-Chem.

[Figure]

Figure R2. Daily variation of nitrate changes due to chemical production (chem) and PBL evolution (vmix) and advection (adv) calculated from WRF-Chem analysis for the Nanjing (a) and Beijing (b) Case, respectively.

[Figure]

Figure R3. Distribution of surface daily mean temperature of east China on (a) March 22nd and (b) April 6th in 2019 and (c) the difference between these two days. (d)(e)(f) are same as above, but for surface ammonia emission rate.

---

## Author Comment (AC3)

**Table R1. Overview of Available Models for Fertilizer Emissions.**

| Reference | Fertilizer | Parameters | Model Type |
|---|---|---|---|
| Fenn and Kissel (1975) | Urea, nitrogen solutions | Time, temperature, application rate | Regression |
| Alkanani and Mackenzie (1992) | Urea, UAN | Temperature, thermodynamic force, wind velocity, soil surface roughness, adsorption and desorption rate constants | Mechanistic |
| Ismail et al. (1991) | Urea solution | Soil temperature, application rate, initial soil moisture content, soil pH, application depth | Regression |
| Kirk and Nye (1991) | Urea | Time, soil moisture content, diffusion factor in soil, vertical distance | Mechanistic |
| Roelle and Aneja (2002) | Hog slurry | Soil temperature | Regression |
| Sogaard et al. (2002) | Cattle and pig Slurry | Soil water content, air temp, wind speed, slurry type, dry matter content of slurry, TAN content of slurry, application method, application rate | Mechanistic |
| Huijsmans et al. (2003) | Slurry | Air temperature, application rate, application method, content of N in slurry, wind speed | Mechanistic |
| Vira et al. (2019) | Fertilizer and livestock waste | Temperature, precipitation, soil moisture, wind speed, spreading of TAN, application rate | Mechanistic |

**Table R2. Sensitivity test setting**

| Experiment | Calculation setting | Sensitivity (%) |
|---|---|---|
| base | $base = CF_{wind} \times CF_{soil_T} \times CF_{soil_m} \times CF_{rain}$ | |
| Sen_temp | $CF_{met1} = CF_{wind} \times (CF_{soil_T} \pm Std_{soilT}) \times CF_{soil_m} \times CF_{rain}$ | $(CF_{met1} - base)/base$ |
| Sen_mois | $CF_{met2} = CF_{wind} \times CF_{soil_T} \times (CF_{soil_m} \pm Std_{soilm}) \times CF_{rain}$ | $(CF_{met2} - base)/base$ |
| Sen_wind | $CF_{met3} = (CF_{wind} \pm Std_{wind}) \times CF_{soil_T} \times CF_{soil_m} \times CF_{rain}$ | $(CF_{met3} - base)/base$ |
| Sen_rain | $CF_{met4} = CF_{wind} \times CF_{soil_T} \times CF_{soil_m} \times (CF_{rain} \pm Std_{rain})$ | $(CF_{met4} - base)/base$ |

**Table R3. Ammonia emissions from different husbandry sources in Tibet and Xinjiang**

| | Free-intensive (Kg/year) | Grazing (Kg/year) |
|---|---|---|
| Tibet | $1.01 \times 10^8$ | $1.12 \times 10^7$ |
| Xinjiang | $2.94 \times 10^8$ | $1.29 \times 10_7$ |

**TableR4. Livestock amount in Tibet and Xinjiang**

| Region | Livestock species | Free-intensive amount (ten thousand) | Grazing amount (ten thousand) |
|---|---|---|---|
| Tibet | Cow and Beef | 421.13 | 262.8 |
| | Sheep and Goat | 981.92 | 632.5 |
| Xinjiang | Cow and Beef | 480.52 | 207.7 |
| | Sheep and Goat | 7351.15 | 1314.5 |

[Figure]

Figure R1. (a) The annual mean emission factor of ammonia in 2019 for eastern China. (b) The sensitivity of emission factors to different meteorological factors.

[Figure]

Figure R2. Illustrations of the application area. The approximate fertilization sequence is indicated by arrows and dates (month-day) (Huo et al., 2015).

[Figure]

Figure R3. (a) Seasonal pattern of monthly averaged near surface temperature in eastern China (18°N–50°N, 95°E–131°E) during 2010-2019. The lower and upper points of vertical lines show the 25th and 75th percentiles, respectively. The black horizontal line represents mean value in 2019. (b) is same as (a) but for mean precipitation.

[Figure]

Figure R4. (a) The annual mean emission factor of ammonia in 2019 for east China; (b) the spatial distribution of basic NH₃ emission in 2019; (c) online NH₃ emission in 2019; (d) traditional MEIC NH₃ emission inventory in 2019.

[Figure]

Figure R5. (a) Piles of manure covered with plastic sheeting; (b)Manure composts in greenhouse.

[Figure]

Figure R6. (a) Slit floor in a farm house; (b)pigs on slit floor; (c) goats on slit floor.

[Figure]

Figure R7. Results of fitting meteorological parameters to ammonia emission fluxes

---

## Author Response (AR1)

**Response to reviewer 1:**

This work develops an online-coupled emission model in WRF-Chem to dynamically describe the ammonia emission rate with varying meteorological and soil conditions. On the basis of this dynamic calculation of ammonia emission and its coupling with regional chemical transport model, it is capable of providing a high-resolution map of ammonia emission across China, and achieves better performance on capturing the spatial pattern and temporal variation on ambient ammonia concentration and also secondary inorganic aerosol. Generally, this work is well designed and structured. The authors have analyzed and elaborated national and regional ammonia emission from agricultural activities in detail. The application of this model is expected to provide a better insight into regional agricultural emission and its role in aerosol formation in China. This topic well fits the scope of the GMD journal and this manuscript is worthy of being published after addressing the issues listed below.

We would like to thank the referee for the encouragements and providing the insightful suggestions, which indeed help us to improve the manuscript. All our responses are provided in line and in color blue.

**Specific comments**

1. This work describes the newly-developed ammonia emission model and its application in China, in which the information and preprocess of input data are introduced in detail. Since that the WRF-Chem model is extensively used across the globe, I personally suggest the authors to provide more descriptions on the data preparation in Section 2 and make the relevant code accessible as well for the convenience of utilization in other regions. Another suggestion is to briefly introduce the WRF-SoilN-Chem model in Section 2.2. Given that the overwhelming majority is discussing the parameterization of ammonia emission rate, the authors need to explain why this model is named SoilN. Are there any other considerations?

**Response:** Accepted. The basic emission data used as static input in Section 2 were mainly obtained from National Bureau of Statistics of China (NBSC, 2020), the source and processing method of the data has been described in detail in previous work by Huang et al. (2012). As suggested by the referee, we added more description of the data

preparation in Section 2. And the relevant code will be accessible by contacting the corresponding author.

For the model name 'SoilN', many studies have demonstrated that soils can emit a variety of nitrogenous gases simultaneously, such as $NH_3$, HONO, and $N_2O$ (Akiyama et al., 2004; Rasool et al., 2019; Liu et al., 2022). At current stage, we have completed the dynamic meteorology-dependent model of $NH_3$ emission, which is the largest part of soil nitrogen emissions. In the future, we plan to develop more soil reactive N gas emission models based on this modeling framework, like HONO or $N_2O$. This is why we call it the "SoilN" model, which have been a comprehensive model involving multiple nitrogen-containing gas emissions from soil.

**Revision:** (Page 5, Line 136-138) "All the basic static emissions data were monthly and were obtained by multiplying the monthly activity data and corresponding static EFs, as shown in equation (1). The province-level activity data of important source were obtained from National Bureau of Statistics of China (NBSC, 2020), the more detailed information of each source could be found in Table S1."

(Page 6, Line 153-156) "The $EF_0$ for urea and ABC were based on experiments carried out in Henan and Jiangsu Province through the micrometeorological method (Cai et al., 1986; Zhu et al., 1989). The $EF_0$ for other less prevalent fertilizers refers to the up-to-date and reliable EFs provided by the European Environment Agency (EEA, 2019), as shown in Table S2. The values of $CF_{pH}, CF_{method}, CF_{rate}$ are all referred to Huang et al. (2012) (Table S2)."

2. The better model performance on reproducing nitrate aerosol and secondary inorganic aerosol on a synoptic scale is a critical improvement of the online calculated NH3 emission scheme. The authors have compared the magnitude of nitrate aerosol in Figure 9. Further analysis ought to be conducted and discussed. For instance, how did the ammonia emission respond to the variations in weather conditions in this case? Is there any regional disparity in the emission sensitivity to a dramatically changing air temperature, and which regions and factors contribute most to the increase in nitrate aerosol?

**Response:** Accepted. As suggested by the referee, we add the results of ammonia emissions in response to weather changes to the manuscript (Figure 8). In Nanjing case, the online emission of ammonia shows dynamic variation and the pattern is quite similar to the temperature and $NH_3$ concentration. However, since the temporal resolution of the offline inventory was only on a monthly basis, the emission intensity showed only monthly differences between March and April, which could not reproduce the variation of daily dramatic emission rate at all. In Beijing case, the variation of online ammonia emission rate was mainly driven by soil moisture variation. It is worth noting that the Online emission rates were higher than Base in both cases and the nitrate and total ammonia better simulated, suggesting that the traditional emission inventories may underestimate soil-emitted ammonia.

We also use diagnostic analysis of WRF-Chem to investigate the causes of nitrate pollution. Figure S4 demonstrates that chemical reaction between $NH_3$ and $HNO_3$ is the main reason for the increase in nitrate in both cases. That is to say, nitrate formation in this region is highly sensitive to the ambient $NH_3$ availability. In 2019 Nanjing case, chemical production and vertical mixing caused the nitrate peaks on Mar 27th and Apr 7th, respectively. The rapid increase during the Beijing case in 2015 was mainly contributed by chemical production. Therefore, the enhanced ammonia emissions in Online model effectively facilitates the neutralization reaction of $NH_3$, which compensates for the underestimated nitrate in Base.

As for the regional disparity in emission, we compare the distribution of ammonia emission rates in east China before (March 22nd) and after (April 6th) the warming. Figure R1 shows that ammonia emissions increased significantly in Jiangsu, Anhui, Henan and the Pearl River Delta regions, but decreased in Hebei and Shandong where temperatures increased by 5-10°C. The decreased emissions in Hebei and Shandong may be due to less soil fertilizer application in April. The increase in ammonia emissions caused by higher temperatures was particularly pronounced in agricultural intensive areas such as Jiangsu and Anhui.

**Revision:** (Page 21, Line 517-522) "In online emission, the $NH_3$ emission flux also shows dynamic variation and the pattern is quite similar to the temperature and $NH_3$ concentration (Fig. 8b). Similar phenomena of a strong relationship between

temperature and ammonia emission has also been proved by previous laboratory experiments (Clay et al., 1990; Pedersen et al., 2021; Niraula et al., 2018). However, since the resolution of the offline inventory was only monthly, the emission intensity showed only monthly differences between March and April, which could not reproduce the variation of daily dramatic emission rate at all."

(Page 21, Line 534-537) "Through the diagnostic analysis of WRF-Chem, the chemical reaction between $NH_3$ and $HNO_3$ was the main reason the nitrate pollution in both cases (Figure S4). That is to say, nitrate formation in this region is highly sensitive to the ambient $NH_3$ availability. In both cases, the online emission rates were higher than base emission and the nitrate and total ammonia better simulated, indicating that the traditional emission inventories may be underestimated."

[Figure]

Figure 8: (a) Time series of total ammonia ($NH_x$) concentrations from 22$^{nd}$ March to 11$^{th}$ April 2019 in Nanjing site. The $T_2$ represents the air temperature at 2m ground level from local site. (b) Daily $NH_3$ emission rate same as (a) in Nanjing site. (c) Time series of $NH_x$ concentrations from 6$^{th}$ to 18$^{th}$ January 2015 in Beijing site. $S_{mois}$ represents the top 5cm thick layer soil water which is derived from WRF model calculations. The mean bias of modelled concentrations is labelled by MB. (d) Daily $NH_3$ emissions rate same as (c) in Beijing site. Blue lines represent the MEIC $NH_3$ emission used in base simulation, red lines represent online emissions from WRF-SoilN-Chem.

[Figure]

Figure S4. Daily variation of nitrate changes due to chemical production (chem) and PBL evolution (vmix) and advection (adv) calculated from WRF-Chem analysis for the Nanjing (a) and Beijing (b) Case, respectively.

[Figure]

Figure R1. Distribution of surface daily mean temperature of east China on (a) March 22nd and (b) April 6th in 2019 and (c) the difference between these two days. (d)(e)(f) are same as above, but for surface ammonia emission rate.

**Technical corrections**

Line 95: I do not think 'aerosols' is necessary since that atmospheric chemistry includes those chemical processes of aerosol.

**Response:** Accepted, we rewrite the sentence.

**Revisions:** (Page 3, Line 97) "WRF-Chem is an extended version of WRF including chemical transformation of trace gases and aerosols simultaneously with meteorology."

Line 111: delete 'environmental' here.

**Response:** Accepted, we remove it.

**Revisions:** (Page 4, Line 114) "The simulated environmental conditions like meteorological element and soil properties provided by WRF solver are transported to $NH_3$ emission model to calculate the meteorology-dependent emission factor (EF)"

Line 113-115: this sentence is difficult to comprehend, which needs to be rephrased.

**Response:** Accepted, we rephrase the sentence.

**Revisions:** (Page 4, Line 116-119) "Consequently, the CF is multiplied by the part (1) basic emission data to obtain meteorology-dependent $NH_3$ emission flux. In Chem section, the flux will be considered as source of $NH_3$ in atmosphere, and participate in the next atmospheric physicochemical processes (deposition, accumulation, convection, boundary layer mixing, and chemistry). At the end of simulation, WRF-SoilN-Chem outputs all meteorological parameters, $NH_3$ emission rates and other chemical diagnostic quantities in WRF's standard format."

Line 123: do the authors mean the static data for emission calculation?

**Response:** Yes, the static data actually refers to the data structure, and the content of which is the basic ammonia emission data. To clarify this, we have rephrased the sentence.

**Revisions:** (Page 5, Line 130-131) "In this study, the basic emission data used as static input were divided into six sections which are fertilizer application, livestock waste, agricultural soil, transport, residential and industry, covering a total of more than 50 emission sources."

Line 242: 'by using WRF-Chem'

**Response:** Accepted. We correct it in the revision.

**Revisions:** (Page 12, Line 301-302) "To evaluate the dynamic $NH_3$ flux model and figure out the aerosol response to dynamic $NH_3$ emission, we designed a pair of parallel experiments by using WRF-Chem."

Line 269: 'largest contributor...'

**Response:** Accepted, we have revised it.

**Revisions:** (Page 12, Line 339) "Regarding livestock waste, the free-range is the largest contributor (65.6 %) to livestock-waste-related $NH_3$ emissions."

**Reference**

Akiyama, H., McTaggart, I. P., Ball, B. C., and Scott, A.: N2O, NO, and NH3 emissions from soil after the application of organic fertilizers, urea and water, Water Air Soil Poll, 156, 113-129, https://doi.org/10.1023/B:Wate.0000036800.20599.46, 2004.

Huang, X., Song, Y., Li, M. M., Li, J. F., Huo, Q., Cai, X. H., Zhu, T., Hu, M., and Zhang, H. S.: A high-resolution ammonia emission inventory in China, Global Biogeochemical Cycles, 26, https://doi.org/10.1029/2011gb004161, 2012.

Liu, Y., Cong, R. H., Liao, S. P., Guo, Q., Li, X. K., Ren, T., Lu, Z. F., and Lu, J. W.: Rapid soil rewetting promotes limited N2O emissions and suppresses NH3 volatilization under urea addition, Environ Res, 212, https://doi.org/10.1016/j.envres.2022.113402, 2022.

National Bureau of Statistics of China (NBSC) (2020), China Statistical Yearbook on Environment 2020 [in Chinese], 248 pp., China Stat. Press, Beijing.

Rasool, Q. Z., Bash, J. O., and Cohan, D. S.: Mechanistic representation of soil nitrogen emissions in the Community Multiscale Air Quality (CMAQ) model v 5.1, Geosci Model Dev, 12, 849-878, https://doi.org/10.5194/gmd-12-849-2019, 2019.

**Response to reviewer 2:**

The manuscript "A dynamic ammonia emission model and the online coupling with WRF-Chem (WRF-SoilN-Chem v1.0): development and evaluation" by Ren et al. describes an ammonia emission model coupled with WRF-Chem that incorporates impacts of meteorological and soil conditions on emission factors (EFs), which aims to provide better estimates of NH3 emissions on both spatial and temporal scales. The modelling results (including fluxes and concentrations) were compared to multiple observations and measurements in China, with evident improvements that can be seen. This study addresses the lack of consideration of environmental impacts on NH3 emissions in current emission inventories. The main assets of this study include: 1) Very simple correction factors were used to represent the climatic dependences of NH3 emissions but covered important processes. 2) The emission model was performed on high spatial and temporal resolution and was coupled to an online climate-chemistry model, which is an advantage for predicting NH3, a short-lived species whose sources can be greatly influenced by meteorological factors. 3) Improvements in estimating NH3 emissions and simulating atmospheric chemistry were made by implementing this emission model. 4) The operation of coupling the emission model with the parent WRF-Chem model looks user-friendly according to the description in the manuscript. Overall, the manuscript is clearly structured and well written. The manuscript should be published in GMD after the authors address the following questions and comments.

We sincerely appreciate your detailed suggestions and comments. These helped to improve this manuscript. The point-by-point responses are listed below.

**Major comments**

1. As mentioned, the emission model is simple but it accounts for important factors that influence NH3 emissions. Therefore, it is crucial to justify why to use the parameterizations shown in the manuscript. The model focuses on China geographically, and most of the corrections for EFs are empirical equations. To what extent these parameterizations can be used globally or in other regions to give reasonable estimates remains unclear. Is it only applicable to China? How well is the model performance in other regions/countries if applying the correction factors in the same way? A useful method would be performing sensitivity tests of some selective parameters in the equations to justify which are the most important ones impacting emissions and what is the relative importance of each parameter. The method used in this study is modifying existing EFs

from emission inventories by including a set of environmental dependencies. Theoretically, this can be done for regions like Europe and US, which could be a future direction.

**Response:** Ammonia emissions from agricultural and livestock farms are influenced by a number of meteorological factors. At the beginning state of the model design, a great deal of literature was extensively investigated and previous models were referred to in order to identify the most important meteorological factors. Based on previous works, Tian et al. (2001) showed that near-surface air temperature, soil moisture, wind speed and precipitation these four meteorological factors had the greatest influence on soil ammonia emissions. Besides, Table S4 illustrates these factors have also been required as important meteorological factors in previous ammonia emission models. Therefore, we chose these four meteorological parameters as the main factors modulating emission rate in the parameterization scheme.

Concerning the model framework, initially we considered two kinds of numerical description; one is the empirical function with meteorological and soil conditions as independent variables, and the other is in the form of a constant differential equation based on soil N balance processes. As you mentioned, the former is overly simple and does not include calculations of soil biological reaction, but it is derived from fitting observational data and the outputs are likely to be more consistent with the measurement and has a lower computational cost. The latter is a mechanistic and process-based model that includes complex biochemical-physical processes such as mineralization, nitrification, denitrification and ionic equilibrium in soil solutions, but the reaction kinetic coefficients for specific processes need to be provided by detailed laboratory data, and bias in the coefficients or imperfect mechanisms can eventually lead to large discrepancies between simulation and observed results. Given the lack of observation and experimental of $NH_3$ flux data, we then decided to use regression model. For the correction coefficients and emission factors in the regression model, we have chosen the experimental results in local Chinese region according to previous research as far as possible (Li et al., 2002; Song et al., 2004; Fan et al., 2006).

Accordingly, the model is currently validated for China region and we have added China to the title for clarity, as you suggested. Since most of the correction coefficients for meteorological factors were derived from local experimental data in China, and also some from Australia, Brazil and Europe, it may also be applicable to

other mid-latitude regions. In our future work, we would like to extend the model development and validation to other agricultural countries in mid-latitudes.

In terms of sensitivity tests of some selective parameters, following your suggestion, we perform some sensitivity experiments to justify which is the most important factor for ammonia emission. In this experiment, we first calculate the average EF value for 2019 as the **Base** value (Figure R1a), and then recalculate the EF by separately changing the values of different meteorological parameters by a range of standard deviations **(Sen_temp/mois/wind/rain)** to test the individual effect of each single meteorological factor on EF. The Table R1 provides the detailed methodology for each experiment. Experiment outputs shown in Figure R1b demonstrates that ammonia emissions are most sensitive to changes in air temperature, followed by soil moisture, wind speed and rainfall in that order, which agrees with the findings of others (Sanchis et al., 2019). We have added more relevant results and discussions in the revised manuscript.

**Revisions:** (Page 7, Line 178-182) "Many dynamically changing meteorological factors have been proved impacts $NH_3$ emissions significantly. Based on previous works, Tian et al. (2001) showed that near-surface air temperature, soil moisture, wind speed and precipitation these four meteorological factors had the greatest influence on soil ammonia emissions. Besides, Table S4 illustrates these factors have also been required as important meteorological factors in previous ammonia emission models. Therefore, we chose these four meteorological parameters as the main factors modulating emission rate in the parameterization scheme."

**Table S4. Overview of Available Models for Fertilizer Emissions.**

| Reference | Fertilizer | Parameters | Model Type |
|---|---|---|---|
| Fenn and Kissel (1975) | Urea, nitrogen solutions | Time, temperature, application rate | Regression |
| Alkanani and Mackenzie (1992) | Urea, UAN | Temperature, thermodynamic force, wind velocity, soil surface roughness, adsorption and desorption rate constants | Mechanistic |

| Ismail et al. (1991) | Urea solution | Soil temperature, application rate, initial soil moisture content, soil pH, application depth | Regression |
|---|---|---|---|
| Kirk and Nye (1991) | Urea | Time, soil moisture content, diffusion factor in soil, vertical distance | Mechanistic |
| Roelle and Aneja (2002) | Hog slurry | Soil temperature | Regression |
| Sogaard et al. (2002) | Cattle and pig Slurry | Soil water content, air temp, wind speed, slurry type, dry matter content of slurry, TAN content of slurry, application method, application rate | Mechanistic |
| Huijsmans et al. (2003) | Slurry | Air temperature, application rate, application method, content of N in slurry, wind speed | Mechanistic |
| Vira et al. (2019) | Fertilizer and livestock waste | Temperature, precipitation, soil moisture, wind speed, spreading of TAN, application rate | Mechanistic |

**Table R1. Sensitivity test setting**

| Experiment | Calculation setting | Sensitivity (%) |
|---|---|---|
| base | $base = CF_{wind} \times CF_{soil_T} \times CF_{soil_m} \times CF_{rain}$ | |
| Sen_temp | $CF_{met1} = CF_{wind} \times (CF_{soil_T} \pm Std_{soilT}) \times CF_{soil_m} \times CF_{rain}$ | $(CF_{met1} - base)/base$ |
| Sen_mois | $CF_{met2} = CF_{wind} \times CF_{soil_T} \times (CF_{soil_m} \pm Std_{soilm}) \times CF_{rain}$ | $(CF_{met2} - base)/base$ |
| Sen_wind | $CF_{met3} = (CF_{wind} \pm Std_{wind}) \times CF_{soil_T} \times CF_{soil_m} \times CF_{rain}$ | $(CF_{met3} - base)/base$ |
| Sen_rain | $CF_{met4} = CF_{wind} \times CF_{soil_T} \times CF_{soil_m} \times (CF_{rain} \pm Std_{rain})$ | $(CF_{met4} - base)/base$ |

[Figure]

Figure R1. (a) The annual mean emission factor of ammonia in 2019 for eastern China. (b) The sensitivity of emission factors to different meteorological factors.

2. The description of running the emission model and the WRF-Chem model in the manuscript is unclear. Did you average the monthly (or annually) basic emission data to obtain the emission with the temporal resolution that is required in the WRF-Chem model? In addition, how did you run the coupled model for China? I assume that you ran nested simulations (I could be wrong). If so, what were the boundary conditions and what emissions did you use for the global run? It is useful to add a section or a paragraph to clarify.

**Response:** We are sorry for the unclear description. The raw resolution of the basic emission data is monthly, which is obtained by multiplying the monthly activity data with the static emission factors for that month. Basic emission data for each month is inputted as volatilizable ammonia into the model at a resolution of at a1 km ×1 km grid.

The simulation of the model is not nested, and only one domain was set. The description of running setting and data used have added in the revised manuscript for clarity.

**Revisions:** (Page 5, Line 138-140) "All the basic static emissions data were monthly and were calculated obtained by multiplying as a product of the monthly activity data and corresponding static EFs, as shown in equation (1)."

(Page 12, Line 308-316) "For 2019, the running time is from Dec 10$^{th}$ 2018 to Dec 31$^{st}$ 2019, each run covered 24 h and the last hour chemical outputs from the preceding

run were used as the initial conditions for the following run. The first 20 days were regarded as the model spin-up period for atmospheric chemistry, so as to better characterize aerosol distributions and minimize the influences of initial conditions and allow the model to reach a state of statistical equilibrium under the applied forcing (Berge et al., 2001). The initial and boundary conditions of meteorological fields were updated from the 6 h NCEP (National Centers for Environmental Prediction) global final analysis (FNL) data with a 1°×1° spatial resolution. NCEP Automated Data Processing (ADP) surface and global upper air observational weather data of wind, temperature and moisture are assimilated to better characterize meteorological factor. The setting of each individual cases is also same as above."

3. Following the above points, a potential weakness I am concerned about is the overall integrity of the emission model. The emission is not calculated from the sources such as the amount of nitrogen in the fertilizers or manure but is derived from the given EFs. However, the EFs are related to the sources such as how much fertilizer nitrogen is applied on land. If you calculated the online EFs at each time step (incorporating the correction factors into the basic emission), the hidden philosophy of the model is that there is a source at every time step which is problematic. Meanwhile, the model might overestimate the emission because the nitrogen reservoirs are depleting. If the basic EFs were assumed to be the same throughout a period (e.g., a month), this does not reflect the tendency of a decreasing emission as there is less and less nitrogen to be emitted. This should be discussed in the manuscript either as an uncertainty evaluation or how the model deals with such a problem

**Response:** Thanks for the suggestion. Fertilizer application is a major source of soil N content and the resultant N emissions in China (Behera et al., 2013; Chen et al., 2022) . During the fertilizer application season, N content of surface soil and ammonia emissions are very intensive and concentrated. So N content in applied fertilizer and application time are crucial to simulate time-varying ammonia emissions. In this model, NH$_3$ emissions do not solely depend on dynamic EF, but also the nitrogen content of soil fertilizers and application time. For example, the activity data ($A_{i,p,m}$) for fertilizer and livestock refers to the nitrogen content of different types of fertilizer and different livestock manures. The monthly amount of fertilizer application is allocated according the application time for tens of different main crops. The

application time related to a crop canopy and the method of application are referenced from the Chinese planting information network (http://www.zzys.gov.cn/) and census data and investigation results (NBSC, 2020; Wang et al., 2008). By this way, the model is capable of characterizing a varying ammonia emission by taking the soil N content increment from fertilizer application into account.

China is one of the largest agricultural countries, the vast area of farmland and the low level of agricultural mechanization make fertilizer application inefficient, often taking half a month or whole month to fertilize. Huo et al. (2015) mentioned that in Hengshui, Hebei, a 2 $km^2$ plot of soil took 18 days to complete fertilizer application (Figure R2). During this long application period, ammonia was emitted sequentially at different application points and peaked at different days, resulting in a relatively flat daily time series. In this paper, although the basic EF is constant in month, emissions will not be significantly overestimated as the daily emission will be relatively flat in the month. However, the issue of temporal losses does exist, the model may partially underestimate the peak ammonia emissions within 1-2 days after fertilizer application. Though the nitrogen depletion is not that obvious on a daily scale, it may introduce some uncertainties in the estimation and we discuss the uncertainty in the revision.

**Revisions:** (Page 25, Line 581-588) "Despite providing more accurate and high-resolution estimation of $NH_3$ emission, the current version of the WRF-SoilN-Chem (version 1.0) still has some limitations including (1) the basic EFs were assumed to be the same throughout the month. However, in reality, the soil pH and nitrogen content of the soil after fertilizer application usually increases rapidly under the hydrolysis of urea and gradually depletes, which leads to variation in EFs as well. So, the constant basic EFs could underestimate the peak emission after fertilization……, (3) the gradual decay of $NH_3$ emissions after fertilization is not added in the model, because the specific fertilizer application date for each agriculture plot is not accessible. The model can be updated and further developed as more laboratory or field measurements data are accessible."

[Figure]

Figure R2. Illustrations of the application area. The approximate fertilization sequence is indicated by arrows and dates (month-day) (Huo et al., 2015).

4. There are some inconsistencies and ambiguities in the overall design of the experimental simulations. It is important to clarify in the manuscript why the year 2019 was chosen to run simulations (rather than a year between 2010 – 2015) and for each comparison what meteorology was used. In the evaluation section, both the annual NH$_3$ concentrations over eastern China by NNDMN (Fig.4) and the site measurements (Fig.8b and Fig.9b) used for model comparisons were not from the year 2019. This raises the question of whether the comparisons are representative enough when using different meteorology for the simulations given that NH$_3$ is strongly influenced by climatic conditions.

**Response:** The reason why we choose 2019 as simulation time period was the consideration of the representativeness of meteorological parameters, data availability and computing resources. The meteorological parameters for 2019 were closed to mean state of the multiple-year average for 2010-2019 (Figure S3), with the difference mostly within the interquartile range. For data reasons, the activity data from 2019 National Bureau of Statistics of China is the most up-to-date and comprehensive and can be used to construct complete static data. Besides, high time-resolved measurements data of ammonia concentrations in Beijing, Nanjing are available for the whole year of 2019. For reasons of computing resource, re-simulating the meteorological factors and ammonia concentrations for 2010–2015 would be computationally intensive and time

consuming. Although the NNDMN data is for 2010-2015, the level of agricultural activity in China has not changed much from 2010 to 2019 and the meteorological factors in 2019 are similar to the mean state in 2010s, so we think we can use 2019 to validate the model.

Besides simulation of 2019, we simulated the cases mentioned in the manuscript separately in this research, including October 2012 (Hengshui flux case), and January 6 - January 18, 2015 (Beijing case). The meteorological fields used are consistent with the timing of the observed cases, so we believe this comparison is representative.

**Revisions:** (Page 19, Line 473-477) "Since the main emission source of atmospheric $NH_3$ and the activity level would not vary a lot in a short time, and the meteorological parameters for 2019 were closed to mean state of the multiple-year average for 2010s (Fig. S3), therefore, a database of atmospheric nitrogen concentration from the nationwide monitoring network (NNDMN) between 2010 to 2015 is used to evaluate the spatial pattern and magnitude of surface $NH_3$ concentrations in China (Xu et al., 2019)."

(Page 12, Line 307-309) "Both two experiments were run for the entire year of 2019 as well as **some individual cases** over the $NH_3$ hot-spot region in eastern China (18° N–50° N, 95° E–131° E) with 20 km grid resolution."

[Figure]

Figure S3. (a) Seasonal pattern of monthly averaged near surface temperature in eastern China (18°N–50°N, 95°E–131°E) during 2010-2019. The lower and upper points of vertical lines show the 25th and 75th percentiles, respectively. The black horizontal line represents mean value in 2019. (b) is same as (a) but for mean precipitation.

5. In Section 3.2, it is unclear whether the description/discussion is for basic $NH_3$ emissions or online $NH_3$ emissions. It is useful to include a map for the online $NH_3$ emissions in Figure 2. Besides, it can be helpful to see the difference between the two.

**Response:** The discussion in Section 3.2 is for online $NH_3$ emissions. As your suggestion, we add the comparison of basic and online emissions and MEIC-$NH_3$ inventory in SI to see the difference among them (Figure S2). Figure 2 in manuscript is the spatial distribution of Basic emission data. However, we may not be able to give an Online version in whole China region because only the eastern China region was simulated with WRF-SoilN-Chem. The remaining ammonia emissions for the west region were

estimated using meteorological data from ERA5 and summed with online outputs to obtain the total emissions for China. Although the ERA5 data is capable of accurately capturing meteorological conditions, it is different from meteorological results of WRF model in terms of resolution and data sources, and thus we add the map of online NH₃ emission in eastern China in the SI.

**Revisions:** (Page 12, Line 303-305) The base simulation used monthly country-level NH₃ inventory based on MEIC NH₃ inventory, which is described by Huang et al. (2012). The comparison of online emissions and fix MEIC NH₃ inventory map are shown in Figure S2.

[Figure]

Figure S2. (a) The annual mean CFs, (b) the spatial distribution of basic NH₃ emission, (c) online NH₃ emission, and (d) traditional MEIC NH₃ emission inventory in 2019 for east China.

6. In Section 3.4, since the model results can be extracted at site locals, such as Nanjing and Beijing, can you present more comparisons in the same way for other sites (as there are over 10 more monitoring sites shown in the boxes in Figure 2)? This can be put into Supplementary

materials. Or it is worth adding a paragraph or a few sentences to clarify why only these two sites were selected for comparisons, i.e., availability, data quality etc.

**Response:** we are sorry for that we did not make it clear in the original manuscript. Only the time-resolved measurements data of ammonia concentrations in Beijing, Nanjing are available for the whole year of 2019. The NNDMN data shown in boxes is multiple-year average for 2010-2015, which cannot to be used to evaluate the temporal variations of model results. Also, we cannot use it to validate the simulation results of seasonal variations for 2019 because the meteorological fields are different. We add a description of these data in the revised manuscript.

**Revisions:** (Page 11, Line 284-289) "Continuous measurements of $NH_3$ and $NH_4^+$ concentration located in Beijing and Nanjing sites of 2019 were used to evaluated the $NH_3$ simulation. In both two sites, the hourly $NH_3$ and $NH_4^+$ were measured by Monitor for Aerosols and Gases in ambient Air (MARGA, MetrohmLtd., Switzerland). In Beijing, the observation is conducted at the Chinese Research Academy of Environmental Sciences (CRAES) (40.05∘ N, 116.42∘ E). In Nanjing, the site is located in the Station for Observing Regional Processes of the Earth System (SORPES) in Nanjing University Xianlin Campus, which is a regional background station in the western part of the YRD region (32.11∘ N, 118.95∘ E) (Ding et al., 2016)."

(Page 11, Line 295-298) "Surface $NH_3$ concentrations in the NNDMN including 43 observation stations were used to compare with simulation. The land types of the NNDMN sites cover cities, farmland, coastal areas, forests and grasslands. Measurements during the period from January 2010 to December 2015 by the NNDMN were used. Surface $NH_3$ concentrations were measured using an active DELTA (DEnuder for Long-Term Atmospheric sampling) (Flechard et al., 2011)."

7. In Section 3.5, I was wondering how you ran the model to get the emissions. First, as mentioned, which year's meteorology was used? Second, did you run the model at the site scale or did you just extract results from the regional simulations for China? How you calculated the basic EFs from the nitrogen application rates given by the field study, i.e, $EF_{0i}, CF_{pH}, CF_{method}$ and $CF_{rate}$ in Equation 2 are not well described. Details can be put in Supplementary Materials.

Third, the field measurement of NH₃ emissions shows relatively comparable magnitudes in terms of daily emissions, i.e., the general daily trend of NH₃ emissions is quite "flat" rather than gradually decreasing over 15 days. The field emissions after fertilization usually reach the maximum within a few days and then decline due to less nitrogen being available for emitting. It is unknown if the model is capable of capturing such features. In other words, the impressive agreement between the modelled emissions and measurements becomes a bit less convincing. It might be due to averaging the monthly emission factor giving the same hourly emission factors, which results in comparable daily sums. Nevertheless, the model reproduces diurnal variations in emissions very well and captures the decreasing feature of NH₃ during a rainy day.

**Response:** In section 3.5, the meteorology field we used is from Oct 11$^{th}$ to Oct 28$^{th}$, coinciding with the time period of the observed fluxes. We have added the detailed value of $EF_{0i}, CF_{pH}, CF_{method}$ and $CF_{rate}$ in the SI.

Indeed, ammonia emissions from soils increase rapidly within a few days after fertilizer application and then gradually decrease and eventually stabilize. In October (the autumn base fertilizer period for wheat in the NCP), winter wheat was applied by spreading fertilizer followed by deep mechanical ploughing without irrigation (Huo et al., 2015). Due to the deep application of the fertilizer and the lack of irrigation, the urea was not fully hydrolyzed, which led to a slow release of ammonia and therefore no significant decline was observed.

The main contribution of this study is the ability to dynamically simulate high resolution ammonia emissions under different meteorological conditions in an air quality model, so the model can characterize ammonia emissions in rainfall weather. The mechanism of gradual decay of ammonia emissions after fertilizer application was not considered in the model for the time being. Although a calculation of the daily decay of NH₃ emission rate was presented by (Huo et al., 2015), it was not added to the model because the specific date of fertilizer application for each farm field isn't accessible. The calculation of the gradual decay of ammonia emissions will be added to a future version when more detailed planting or fertilizing date data is available.

**Revisions:** (Page 6, Line 156-159) "The EF₀ for urea and ABC were based on experiments carried out in Henan and Jiangsu Province through the micrometeorological method (Cai et al., 1986; Zhu et al., 1989). The EF₀ for other less prevalent fertilizers refers to the up-to-date and reliable EFs provided by the European Environment Agency

(EEA, 2019), as shown in Table S2. The values of $CF_{pH}, CF_{method}, CF_{rate}$ are all referred to Huang et al. (2012) (Table S2)."

**Table S2. EFs, Expressed as Percentage of Volatilized NH₃-N From Applied Fertilizer-N, and Static Correction Coefficients for Different N Fertilizer Categories.**

| Fertilizer Categories | Measured EFs | | Coerrection Coefficients | |
|---|---|---|---|---|
| | Acid Soil | Alkaline Soil | $CF_{rate}$ [a] | $CF_{method}$ [b] |
| Urea | 8.8[c] | 30.1[c] | 1.18 | 0.32 |
| ABC | 18.2[d] | 39.1[d] | 1.18 | 0.32 |
| AN | 1.6[e] | 3.3[e] | 1.18 | 0.32 |
| Others | 1.4[e] | 2.0[e] | 1.18 | 0.32 |

[a]Values are derived from *Li et al. (2002), Song et al. (2004), and Fan et al. (2006)*.
[b]Values are derived from *Lu et al. (1980), Qu (1980), Fillery et al. (1986), Zhang and Zhu (1992), Li and Ma (1993), and Cai et al. (2002)*.
[c]Measurement results of *Zhu et al. (1989)*.
[d]Measurement results of *Cai et al. (1986)*.
[e]Values recommended by EEA (2019).

8. There is a lack of discussion on uncertainty from the new emission model. This can be either some text descriptions discussing various uncertainties, or any back-of-envelope calculations derived from the potential sensitivity tests.

**Response:** Accepted, we add the discussion on uncertainty of the model to Section 4. "Emission uncertainty is associated with static EF and dynamic correction factor. The EFs should be related to the sources such as how much nitrogen remains in soil. In reality, the EF and nitrogen content of the soil after fertilizer application usually increases rapidly under the hydrolysis of urea and gradually depletes. However, the basic EFs were assumed to be the same throughout the period, which could underestimate the peak emission after fertilization. Besides, the model assumes that soil pH is constant on a monthly scale, but in reality, after soil application of nitrogen fertilizer, soil pH increases for a few days and then decreases to its original state. The static basic EFs and soil pH may lead to the model underestimate the peak after fertilizer application.

**Revisions:** (Page 25, Line 584-596) "the current version of the WRF-SoilN-Chem (version 1.0) still has some limitations including …… (2) the meteorological CF parameterization scheme used in the model same for all agricultural soil. However, the emissions can be different from soils with the same water content but different porosity (soil water

content at saturation), which is not considered in the model. ......The model can be updated and further developed as more laboratory or field measurements data are accessible."

9. Although WRF-Chem can be run globally, the study focuses on China. I would suggest revising the title of the manuscript to be more specific on its spatial coverage. For example: "A dynamic ammonia emission model and the online coupling with WRF-Chem (WRF-SoilN-Chem v1.0): development and regional evaluation in China" or "A dynamic ammonia emission model for China and the online coupling with WRF-Chem (WRF-SoilN-Chem v1.0): development and evaluation".

**Response:** Accepted, we revise the title to "A dynamic ammonia emission model and the online coupling with WRF-Chem (WRF-SoilN-Chem v1.0): development and regional evaluation in China"

10. I would personally suggest putting Section 3.5 in front of Section 3.3. Section 3.5 directly compared modelled $NH_3$ emissions to measured emissions, while Sections 3.3 and 3.4 are comparisons between modelled and observed/measured atmospheric $NH_3$ concentrations. By doing that, the evaluation is in the order of "emission, $NH_3$ concentrations, and aerosol concentrations". This point is optional.

**Response:** Thanks for the suggestion, we accept it. We restructure Section 3.3 and 3.5 in the revision.

**Specific comments**

P6L138: This is not entirely correct because soil pH can increase after urea application. Urea hydrolysis consumes H+ ions so it tends to lead to more NH3 emissions. The extent of pH increase is dependent on the soil's pH buffering capacity. Although this is complex, it should be specifically clarified that the model assumes constant soil pH. Please modify "In the fertilizer a pplication section, soil pH … are relatively stable in a short time."

**Response:** Accepted. Thanks for this suggestion. Indeed, the soil pH can increase significantly after urea application or livestock urine deposition (Curtin et al., 2020), but some lab experiments shown that with time, the soil pH can gradually fall back to its original state in 30 days due to the nitrification of $NH_4^+$ resulting the production of acid, thus the soil pH are relatively stable on monthly scale. Due to the complexity of soil pH measurements and simulation, only the HWSD database currently provides soil pH data with low temporal resolution (annual resolution), and it is also difficult for current models to provide high temporal resolution simulations of soil pH. Therefore, the soil pH is assumed to be relatively stable in short time. We modify the sentence "In the fertilizer application section, soil pH … **are assumed to be** relatively stable in a short time." to "In the fertilizer application section, depends on the habits of the farmers, the fertilizer application rate and method are relatively stable in a short time, thus they were introduced as stabile parameters $EF_{static_{fertilizer}} = EF_{0i} \times CF_{pH} \times CF_{method} \times CF_{rate}$ to adjust emission factors for static conditions. As for the soil pH, the actually it can increase after urea application or livestock urine deposition (Curtin et al., 2020), but it will gradually fall back to its original state in 30-50 days due to the nitrification of $NH_4^+$ resulting the production of acid. Besides, due to the complexity of soil pH measurements and simulation, it is difficult to obtain observation or simulation data with high temporal resolution. So, in this model, the soil pH is assumed to be a stable parameter."

**Revisions:** (Page 6, Line 146-151) "In the fertilizer application section, the fertilizer type, soil pH, fertilizer application rate and method are introduced as parameters to develop EFs for specific conditions. The fertilization rate and method are relatively stable in a month based on the farmers' traditional growing habits. As for soil pH, although it significantly increases after fertilizer application, it gradually fall back to normal state within 30 days due to the nitrification of $NH_4^+$ (Curtin et al., 2020). Thus, the pH, fertilizer rate and method were assumed to be relatively stable in monthly scale and were introduced as stable parameters to adjust emission factors for static conditions."

P6L164-165: Same comments as mentioned above and in the 3rd points from Major comments. Ammoniacal N concentrations can vary greatly throughout the application period. Ammoniacal N is lost from the soils through various pathways such as volatilization, nitrification, and physical

transport like runoff and diffusions, which can affect the concentration of ammoniacal N. Ammoniacal N also exists in different phases, e.g., can be adsorbed on soil particles, which depends upon soil moisture and other factors. Since the model uses simple correction factors and does not include detailed soil processes, it should clearly state the assumptions to avoid misleading. Please consider rephrasing into "…ammoniacal N concentration and soil pH **are assumed to be** stable in short time …"

**Response:** Accepted. The sentence is modified as the referee suggested.

**Revisions:** (Page 6, Line 149-150) "Thus, the pH, fertilizer rate and method were assumed to be relatively stable in monthly scale and were introduced as stable parameters to adjust emission factors for static conditions."

P7L177: Is there any statistical or activity data for China supporting that "the handle of excrement is usually settled in closed containers"?

**Response:** Jia Wei (2014) found that for traditional farmer farming, 37% of the manure will be used for composting, 11% will be used for biogas production, and 34% will be used for returning to the field to be exposed to air. Biogas is often placed in sealed tanks due to the need for an anaerobic environment. As for composting, according to the technical specifications of livestock and poultry manure composting in China's agricultural industry standards (Ministry of Agriculture and Rural Affairs, 2019), composting site should be " anti-seepage, rain-proof, anti-spillage", so people usually lay fine soil and straw on the ground, and spread a layer of mud or plastic sheeting on the manure or just compost the manure in a closed greenhouse (Figure R3), so as to form a closed environment to avoid the influence of external temperature, wind speed and precipitation. We rewrite the "closed containers" to "closed environment" in the revision for clarity.

[Figure]

Figure R3. (a) Chicken farm manure disposal covered with plastic sheeting (2023); (b)Manure composts in greenhouse (2014).

**Revisions:** (Page 7, Line 200-205) "As for the manure storage section, 77% of the manure will be used for composting, 23% will be used for biogas production (Jia, 2014). Biogas is often placed in sealed tanks due to the need for an anaerobic environment. As for composting, the handle of manure site should be "anti-seepage, rain-proof, anti-spillage" (Ministry of Agriculture and Rural Affairs, 2019), so people usually lay fine soil and straw on the ground, and spread a layer of mud or plastic sheeting on the manure or just compost the manure in a closed greenhouse, so as to form a closed environment to avoid the influence of external temperature, wind speed and precipitation."

P7L192: Please explain why soil moisture correction is applied for modelling emissions from animal houses. There are houses with concrete floors or slatted floors. Or is this equation only applied to specific management

**Response:** For intensive farming, to keep the house clean and facilitate machine cleaning, the floor made of cement is often with holes or slits to allow the leakage of manure onto the soil below (Figure R4), which is easy to scrape off using soil-turning machines. So, we believe that manure will still be in touch with the soil and therefore will be affected by surface soil moisture.

[Figure]

Figure R4. (a) Slit floor in a pig farm house (2016); (b) pigs on slit floor (2020); (c) Macro goat dairy in China (2017).

**Revisions:** (Page 8, Line 219-221) "To keep the house clean and animals comfortable, the floor of farmhouse is often with holes or slits to allow the leakage of manure onto the soil below, making the manure easy to be swept away by cleaning machines. So, the manure is still in touch with the soil and therefore will be affected by surface soil moisture."

P8L211-216: It is useful to include more details for the experimental study carried out by your research group which is used for deriving soil temperature correction factor, i.e., what method was used? How was the study designed? Any reference?

**Response:** Based on the Gibbs free energy equation and the form of correction factor proposed by Gyldenkaerne et al. (2005), the effect of soil temperature on ammonia emission should be exponential (Roelle and Aneja, 2005). And since ammonia emission is influenced by surface temperature and soil temperature gradient, we consider the correction factor equation for soil as equation (1) and (2). Then we combined the activity level data above and multiple factors of temperature, wind speed and precipitation and soil moisture to obtain an equation with coefficients that were fitted to the flux data in April to obtain the coefficients in equation (3).

$$CF_{soil_T} = CF_{soilT-gradient} \times CF_{soilT-surface} \tag{1}$$

$$CF_{soil_T} = e^{(a1 \times \Delta soil_T + b1)} \times e^{(a2 \times soilT + b2)} \tag{2}$$

$$CF_{soil_T} = e^{(0.093 \times \Delta soil_T - 0.97 + 0.018 \times soil_T)} \tag{3}$$

**Revisions:** The above description in response are added in SI.

[Figure]

Figure S1. Results of fitting meteorological parameters to ammonia emission fluxes

P8L225-229: It is worth mentioning in the manuscript that emissions can be different from soils with the same water content but different porosity (soil water content at saturation), which is not considered in the model.

**Response:** Accepted, thanks for the suggestion. We mention this in the uncertainty discussion in section 4.

**Revisions:** (Page 25, Line 584-586) "the current version of the WRF-SoilN-Chem (version 1.0) still has some limitations including …… (2) the meteorological CF parameterization scheme used in the model same for all agricultural soil. However, the emissions can be different from soils with the same water content but different porosity (soil water content at saturation), which is not considered in the model……."

P9L230-235: I was wondering what is the underlying mechanism that rainfall affects NH3 emissions in the model. Is it because of infiltration, runoff, or changes in soil moisture? Is there a double-counting issue here between the rainfall and soil moisture correction?

**Response:** As we know, ammonia readily dissolves in water. The maximum concentration of ammonia in water (a saturated solution) has a density of 0.880 g/cm3. Some studies proved that ammonia in the atmosphere can be captured by raindrops (Delitsky and Baines, 2016; Shimshock and De Pena, 1989). We suggest the underlying mechanism is the scavenging of ammonia fluxes by raindrops near the surface.

In equation (10) for rainfall, the amount of ammonia emitted into the air decreases as the rainfall increases. However, in equation (9) for soil moisture, soil wetting promotes ammonia emissions. We therefore believe that there is no significant double-counting of the effects of rainfall and soil moisture. We add the underlying mechanism of rainfall in the manuscript.

**Revisions:** (Page 9, Line 261-262) "As $NH_3$ readily dissolves in water, $NH_3$ flux can be scavenged by raindrops near the surface (Delitsky and Baines, 2016; Shimshock and De Pena, 1989). Several studies reported that rainfall events after fertilizer application can influence the maximum potential emission of $NH_3$ in the field (Parker et al., 2005; Smith et al., 2009)."

P11L268: Again, since urea is the most widely used fertilizer in China, a discussion of the uncertainty caused by not including soil pH change is missing.

**Response:** (Page 25, Line 581-584) Accepted, the uncertainty caused by the soil pH is discussed in Section 4.

**Revisions:** (Page 25, Line 581-584) "the basic EFs were assumed to be the same throughout the month. However, in reality, the soil pH and nitrogen content of the soil after fertilizer application usually increases rapidly under the hydrolysis of urea and gradually depletes, which leads to variation in EFs as well. So, the constant basic EFs could underestimate the peak emission after fertilization."

P13L293-294: Tibet has very little NH3 emissions as shown in Figure 2, while there are some hot spots in Xinjiang province. Meanwhile, sheep are not a significant contributor to NH3 emissions especially when grazing is dominant. What does the model tell? You should be able to diagnose the sectoral emissions from the model.

**Response:** After verification of activity data and model results, we found that in the Xinjiang region, free-range farming and intensive farming are the main contributors to ammonia emissions (Table R2). Among them, the number of goats and sheep farmed is much higher than that of beef cattle and dairy cows (Table R3). Due to the high

altitude and sparse population, livestock farming is not well developed in the Tibetan region and ammonia emissions are not high.

**Revisions:** (Page 15, Line 363-364) We amend the original statement to "In Xinjiang provinces, sheep are widely raised, which is responsible for remarkable ammonia emissions related to sheep manure management."

**Table R2. Ammonia emissions from different husbandry sources in Tibet and Xinjiang**

|  | Free-intensive (Kg/year) | Grazing (Kg/year) |
|---|---|---|
| Tibet | $1.01 \times 10^8$ | $1.12 \times 10^7$ |
| Xinjiang | $2.94 \times 10^8$ | $1.29 \times 10_7$ |

**Table R3. Livestock amount in Tibet and Xinjiang**

| Region | Livestock species | Free-intensive amount (ten thousand) | Grazing amount (ten thousand) |
|---|---|---|---|
| Tibet | Cow and Beef | 421.13 | 262.8 |
|  | Sheep and Goat | 981.92 | 632.5 |
| Xinjiang | Cow and Beef | 480.52 | 207.7 |
|  | Sheep and Goat | 7351.15 | 1314.5 |

P14L337: Then why not simulate 2010-2015?

**Response:** As mentioned in our response to Q4, we chose to simulate 2019 based on the meteorological representation and data availability for this year. As for why we did not simulate 2010-2015, one reason is that re-simulating the meteorological factors and ammonia concentrations for 6 years would be computationally intensive and time consuming, and another reason is that the NNDMN data has a low temporal resolution and can only be used to validate spatial simulation effects, which can also be done with satellite data (IASI). Therefore, we do not consider it necessary to re-simulate 2010-2015 after having already simulated and compared the 2019 ammonia data.

P19L431-432: As stated in the manuscript, there are no diurnal variations in the inventory. However, some diurnal variation can be seen as shown in Figure 7b from the base run (e.g., 10:00 to 18:00 has higher emission than other times), which is confusing.

**Response:** The time resolution of MEIC model is monthly. However, WRF-Chem requires the hourly input emission data. To integrate the MEIC ammonia inventory into WRF-Chem simulations, we adopted a diurnal profile with 80% of the $NH_3$ emissions in the daytime, following previous studies (Zhu et al., 2015; Asman, 2001; Du et al., 2020). We clarify this in the revision.

**Revisions:** (Page 18, Line 437-440) "However, fixed inventory used in the base simulation are monthly and has no diurnal variation of emission. To integrate this inventory into WRF-Chem simulation, we adopted a diurnal profile with 80 % of $NH_3$ emissions in the daytime, following previous studies (Du et al., 2020)"

**Other comments**

P2L63: "environment elements" to "environmental elements".

**Response:** Accepted, we modify it as suggested.

**Revisions:** (Page 2, Line 64) "Generally, the environmental elements appreciably influencing ammonia emissions ……"

P5L124: "agriculture soil" to "agricultural soil".

**Response:** Accepted, we correct the word.

**Revisions:** (Page 5, Line 130-131) "…… static input were divided into six sections which are fertilizer application, livestock waste, agricultural soil……"

P9L235: Numbering of the equation is missing.

**Response:** Accepted, the number label is added now.

**Revisions:** (Page 9, Line 267) $CF_{rain} = 1/(3.2 \times rainfall + 1)$ (10)

P18: Labelling of Figure 7 is missing.

**Response:** Accepted, we add it, as shown below.

[Figure]

Figure 4: (a) Time series of observed (black symbol) and WRF-Chem Online model NH$_3$ flux (red line) above China Hengshui agri-field from the 11[th] to 27[th] October 2012. (b) Diel hourly box plots of observations flux measurements (grey), paired with online model results (pink) and base model results (blue). The 5[th] and 95[th] percentiles are represented by the whiskers, the 25th and 75th quantiles are enclosed in the box, the median is represented by the horizontal line through the box, and mean value is the dot in the box. Diurnal profile of emissions from agriculture is applied in base experiment following Du et al. (2020)

P18: Consider using grams or kilograms rather than mol.

**Response:** Accepted. We modify the units as shown in above figure.

**Reference**

Alkanani, T. and Mackenzie, A. F.: Effect of Tillage Practices and Hay Straw on Ammonia Volatilization from Nitrogen-Fertilizer Solutions, Can J Soil Sci, 72, 145-157, https://doi.org/10.4141/cjss92-014, 1992.

Asman, W. A. H.: Modelling the atmospheric transport and deposition of ammonia and ammonium: an overview with special reference to Denmark, Atmos Environ, 35, 1969-1983, https://doi.org/10.1016/S1352-2310(00)00548-3, 2001.

Behera, S. N., Sharma, M., Aneja, V. P., and Balasubramanian, R.: Ammonia in the atmosphere: a review on emission sources, atmospheric chemistry and deposition on terrestrial bodies, Environ Sci Pollut R, 20, 8092-8131, https://doi.org/10.1007/s11356-013-2051-9, 2013.

Chen, Y. F., Zhang, L., Zhao, Y. H., Zhang, L. J., Zhang, J. W., Liu, M. Y., Zhou, M., and Luo, B.: High-Resolution Ammonia Emissions from Nitrogen Fertilizer Application in China during 2005-2020, Atmosphere-Basel, 13, https://doi.org/10.3390/atmos13081297, 2022.

Chicken farm manure disposal covered with plastic sheeting: https://new.qq.com/rain/a/20200211A0AK4I00?pc, last access: 15 February 2023.

Curtin, D., Peterson, M. E., Qiu, W., and Fraser, P. M.: Predicting soil pH changes in response to application of urea and sheep urine, J Environ Qual, 49, 1445-1452, https://doi.org/10.1002/jeq2.20130, 2020.

Delitsky, M. L. and Baines, K.: Scavenging of ammonia by raindrops in Saturn's great storm clouds, October 01, 2016.

Du, Q. Y., Zhao, C., Zhang, M. S., Dong, X., Chen, Y., Liu, Z., Hu, Z. Y., Zhang, Q., Li, Y. B., Yuan, R. M., and Miao, S. G.: Modeling diurnal variation of surface PM2.5 concentrations over East China with WRF-Chem: impacts from boundary-layer mixing and anthropogenic emission, Atmos Chem Phys, 20, 2839-2863, https://doi.org/10.5194/acp-20-2839-2020, 2020.

Fan, X. H., Y. S. Song, D. X. Lin, L. Z. Yang, and J. F. Luo: Ammonia volatilization losses and N-15 balance from urea applied to rice on a paddy soil [in Chinese], J. Environ. Sci., 18(2), 299–303. https://doi.org/ CNKI:SUN:HJKB.0.2006-02-016. 2006.

Fenn, L. B. and Kissel, D. E.: Ammonia Volatilization from Surface Applications of Ammonium-Compounds on Calcareous Soils .4. Effect of Calcium-Carbonate Content, Soil Sci Soc Am J, 39, 631-633, https://doi.org/ 10.2136/sssaj1975.03615995003900040019x, 1975.

Gyldenkaerne, S., Skjoth, C. A., Hertel, O., and Ellermann, T.: A dynamical ammonia emission parameterization for use in air pollution models, J Geophys Res-Atmos, 110, https://doi.org/10.1029/2004jd005459, 2005.

Huang, X., Song, Y., Li, M. M., Li, J. F., Huo, Q., Cai, X. H., Zhu, T., Hu, M., and Zhang, H. S.: A high-resolution ammonia emission inventory in China, Global. Biogeo. Chem., Cy., 26, https://doi.org/10.1029/2011gb004161, 2012.

Huijsmans, J. F. M., Hol, J. M. G., and Vermeulen, G. D.: Effect of application method, manure characteristics, weather and field conditions on ammonia volatilization from manure applied to arable land, Atmos Environ, 37, 3669-3680, https://doi.org/10.1016/S1352-2310(03)00450-3, 2003.

Huo, Q., Cai, X. H., Kang, L., Zhang, H. S., Song, Y., and Zhu, T.: Estimating ammonia emissions from a winter wheat cropland in North China Plain with field experiments and inverse dispersion modeling, Atmos Environ, 104, 1-10, https://doi.org/10.1016/j.atmosenv.2015.01.003, 2015.

Ismail, K. M., Wheaton, F. W., Douglass, L. W., and Potts, W.: Modeling Ammonia Volatilization from Loamy Sand Soil Treated with Liquid Urea (Correction of Transactions of the Asae, Vol 34, No 3, Pg 756, 1991), T Asae, 34, 756-763, 1991.

Jia, W.: Studies on the Evaluation of Nutrient Resources Derived from Manure and Optimized Utilization in Arable Land of China, China Agricultural University, 2014.

Kirk, G. J. D. and Nye, P. H.: A Model of Ammonia Volatilization from Applied Urea .5. The Effects of Steady-State Drainage and Evaporation, J Soil Sci, 42, 103-113, https://doi.org/10.1111/j.1365-2389.1991.tb00095.x, 1991.

Li, G., B. Li, and D. Chen: Ammonia volatilization from large field planted with winter wheat and summer maize [in Chinese], Acta Agric. Boreali Sin., 17(1), 76–81, 2002.

Macro goat dairy in China: https://www.iga-goatworld.com/blog/situation-of-dairy-goats-in-the-world, last access: 15 February 2023, 2017.

Manure composts in greenhouse: http://www.gzwybio.com/sea_show_249.html, last access: 15 February 2023, 2014.

National Bureau of Statistics of China (NBSC) (2020), China Statistical Yearbook on Environment 2020 [in Chinese], 248 pp., China Stat. Press, Beijing.

Pigs on slit floor: https://thesamikhsya.com/breaking-news/amid-covid-19-african-swine-fever-kills-over-13000-pigs-in-assam, last access: 15 February 2023, 2020.

Roelle, P. A. and Aneja, V. P.: Characterization of ammonia emissions from soils in the upper coastal plain, North Carolina, Atmos Environ, 36, 1087-1097, https://doi.org/10.1016/S1352-2310(01)00355-7, 2002.

Roelle, P. A. and Aneja, V. P.: Modeling of ammonia emissions from soils, Environ Eng Sci, 22, 58-72, https://doi.org/10.1089/ees.2005.22.58, 2005.

Sanchis, E., Calvet, S., del Prado, A., and Estelles, F.: A meta-analysis of environmental factor effects on ammonia emissions from dairy cattle houses, Biosyst Eng, 178, 176-183, https://doi.org/10.1016/j.biosystemseng.2018.11.017, 2019.

Shimshock, J. P. and De Pena, R. G.: Below-cloud scavenging of tropospheric ammonia, Tellus B, 41, 296-304, https://doi.org/10.1111/j.1600-0889.1989.tb00308.x, 1989.

Slit floor in a pig farm house: http://yzxy.nxin.com/html/20160618/41885.html, last access: 14 February 2023, 2016.

Sogaard, H. T., Sommer, S. G., Hutchings, N. J., Huijsmans, J. F. M., Bussink, D. W., and Nicholson, F.: Ammonia volatilization from field-applied animal slurry - the ALFAM model, Atmos Environ, 36, 3309-3319, https://doi.org/10.1016/S1352-2310(02)00300-X, 2002.

Song, Y., X. Fan, and D. Lin: Ammonia volatilation from paddy fields in the Taihu lake region and its influencing factors [in Chinese], Acta Pedol. Sin., 41(2), 265–269. https://doi.org/CNKI:SUN:TRXB.0.2004-02-015, 2004.

Tian, G. M., Cai, Z. C., Cao, J. L., and Li, X. P.: Factors affecting ammonia volatilisation from a rice-wheat rotation system, Chemosphere, 42, 123-129, https://doi.org/10.1016/S0045-6535(00)00117-X, 2001.

Vira, J., Hess, P., Melkonian, J., and Wieder, W. R.: An improved mechanistic model for ammonia volatilization in Earth system models: Flow of Agricultural Nitrogen version 2 (FANv2), Geosci Model Dev, 13, 4459-4490, https://doi.org/10.5194/gmd-13-4459-2020, 2019.

Wang, J., W. Ma, and R. Jiang: Analysis about amount and ratio of basal fertilizer and topdressing fertilizer on rice, wheat, maize in China [in Chinese], Chin. J. Soil Sci., 39(2), 329–333. https://doi.org/10.19336/j.cnki.trtb.2008.02.024, 2008

Zhu, L., Henze, D., Bash, J., Jeong, G. R., Cady-Pereira, K., Shephard, M., Luo, M., Paulot, F., and Capps, S.: Global evaluation of ammonia bidirectional exchange and livestock diurnal variation schemes, Atmos Chem Phys, 15, 12823-12843, https://doi.org/10.5194/acp-15-12823-2015, 2015.